# Evaluating Numerical Reasoning in Text-to-Image Models

**Ivana Kajić**
Google DeepMind

**Olivia Wiles**
Google DeepMind

**Isabela Albuquerque**
Google DeepMind

**Matthias Bauer**
Google DeepMind

**Su Wang**
Google DeepMind

**Jordi Pont-Tuset**
Google DeepMind

**Aida Nematzadeh**
Google DeepMind

## Abstract

Text-to-image generative models are capable of producing high-quality images that often faithfully depict concepts described using natural language. In this work, we comprehensively evaluate a range of text-to-image models on numerical reasoning tasks of varying difficulty, and show that even the most advanced models have only rudimentary numerical skills. Specifically, their ability to correctly generate an exact number of objects in an image is limited to small numbers, it is highly dependent on the context the number term appears in, and it deteriorates quickly with each successive number. We also demonstrate that models have poor understanding of linguistic quantifiers (such as "a few" or "as many as"), the concept of zero, and struggle with more advanced concepts such as partial quantities and fractional representations. We bundle prompts, generated images and human annotations into GECKONUM, a novel benchmark for evaluation of numerical reasoning.

## 1 Introduction

Recent generative text-to-image models can produce images of impressive quality in a variety of styles and following the text descriptions provided by users [4, 23, 31]. However, they may still fail to accurately generate images where the given descriptions contain numbers and quantities (such as "7 pistachios", as shown in Figure 1) [20, 22]. While recent work has focused on designing metrics, benchmarks and methods for evaluating specific capabilities of text-to-image generative models, such as alignment [33, 15, 9], compositionality [16], or spatial reasoning [13], there is no comprehensive benchmark for evaluation of numerical reasoning.

We address this gap by proposing GECKONUM, a comprehensive and controlled benchmark of text prompts aimed at evaluating different aspects of numerical reasoning in text-to-image models. We formalize evaluation as three tasks: exact number generation, approximate number generation, and reasoning about partial quantities. For each task, we design various template types to control for different variables such as sentence structure, the context in which the number words occur in, and the number of attributes/entities in a prompt. Table 1 shows examples of numerical tasks and the associated prompt types.

Using GECKONUM, we evaluate twelve models chosen from five different model families (DALL·E 3, Midjourney, Imagen, Muse and Stable Diffusion): we generate images for these models and collect human annotations to measure whether the images correctly match the prompts with respect to numerical reasoning. Our benchmark consists of 1386 text prompts, 52,721 generated images, and a total of 479,570 human annotations that we release.[1]

---

[1] https://github.com/google-deepmind/geckonum_benchmark_t2i

38th Conference on Neural Information Processing Systems (NeurIPS 2024) Track on Datasets and Benchmarks.

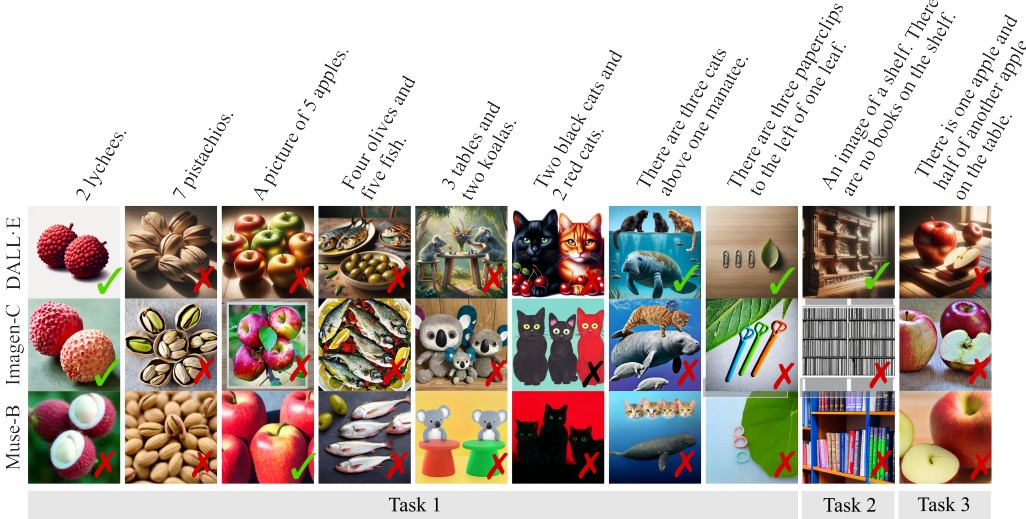

Figure 1: Examples of images generated by selected models: DALL·E 3, Imagen-C and Muse-B. Correctly generated images are marked with a check mark "✓", and incorrect with a cross mark "✗".

Overall, our results demonstrate that the recent generative text-to-image models have rudimentary numerical reasoning skills, and are most accurate when tested for generation of small exact quantities. We highlight the utility of GECKONUM as an evaluation benchmark: it can discriminate between models, even the powerful ones, such as Imagen-D [31] and DALL·E 3 [2], that are similar in terms of high image quality. Finally, we demonstrate that our benchmark can be used to drive progress in related research areas, such as the development of automatic evaluation metrics and evaluation and improvement of pretrained vision–language models on counting.

## 2 Related Work

**Text-to-image benchmarking.** Benchmarks used for evaluation of text-to-image models typically consist of a set of text prompts that target a specific capability. Some benchmarks, such as DALL·E 3 Eval [2] are more general and aim to capture real world use cases, while others contain a variety of challenges, such as prompts in DrawBench [25], PartiPrompts [36], and HEIM [17]. Others are developed to interrogate models on more specific capabilities such as alignment (e.g. TIFA [15], DSG [9] and Gecko [33]) or compositionality (e.g. T2I-CompBench [16]).

Often, prompts in such benchmarks are curated from the data harvested from the Web, and they may include numbers or other numerical concepts relevant to evaluation of numerical reasoning. However, the number of such prompts in existing benchmarks is often limited, and the complexity of prompts may not be appropriate to evaluate models on numerical reasoning as a correct machine interpretation of such prompts often requires a combination of several different reasoning skills. For example, the understanding of the number "2" in a simple prompt such as *"Two zebras in Cape Town"* (from CountBench [20]) also requires an interpretation of a geographical location. For the correct interpretation of *"Two dogs in a grassy field with one dog holding an orange disc."* (from MS COCO [6]) the model would need to correctly generate several objects (i.e. *"dog", "disc"*), relations between them (*"holding"*), and correctly bind attributes (*"orange"*) to objects. While the complexity of such prompts may be representative of the complexity observed in natural language, it may hinder our ability to accurately evaluate numerical reasoning in text-to-image models.

The prompt set in GECKONUM is vastly more comprehensive and it systematically covers various dimensions of evaluation that affect numerical reasoning, such as different number ranges, types of nouns, different ways of representing numbers, approximations based on linguistic quantifiers, and reasoning about partial quantities that are missing in other datasets. We show that to evaluate a specific capability thoroughly requires an extensive, comprehensive dataset. Most similar to our work is [22], which investigates number generation for text-to-image models. While [31] show

that number generation improves with larger model sizes, their set of prompts is small (N=59) and focuses only on the simplest prompt structure. We go beyond this by evaluating a comprehensive set of model families, considering other properties of numerical reasoning such as estimation and conceptual quantitative reasoning, and a more comprehensive breakdown of performance based on how commonplace an entity is.

**Measuring counting in image-to-text models.** While numerical reasoning has no standardized benchmark for text-to-image generation, there are some widely accepted benchmarks within the visual question answering domain. For example, TallyQA [1] and CountBench [20] both measure the counting ability of question-answering models. CountBench is a small dataset of only 540 images whereas TallyQA contains approximately 20K images for evaluation. However, despite TallyQA's size, the distribution of numbers is highly skewed towards small numbers (e.g. 1, 2); and the quality of the images and associated labels is mixed.

# 3 Tasks to Examine Numerical Reasoning

We evaluate text-to-image models on different aspects of numerical reasoning formulated as three numerical tasks and a set of text prompt types for each task, spanning different levels of difficulty. Our working definition of numerical reasoning draws inspiration from literature in numerical cognition, and concerns both the ability to reason in abstract terms and the ability to manipulate such abstractions across different contexts. Specifically, one fundamental aspect of abstract reasoning is the ability to form a representation of a set size independently of the identity of objects, known as *The abstraction principle* [11]. For example, understanding that "two" refers to the same quantity in "Two apples" as it does in "Two letters" even though apples and letters are different objects. We first define the tasks and describe what aspect of numerical reasoning they intend to evaluate, followed by description of prompts that aim to require those aspects of reasoning. These prompts are generated using intentionally designed 12 templates where we sweep over combinations of numbers and selected word nouns, resulting in 1386 prompts in total. The 12 prompt types, including prompts and example templates used to generate the prompts are shown in Table 1, with further details available in Table A2.

## 3.1 Task 1: Exact Number Generation

Task 1 examines a model's ability to correctly generate an exact number of objects. We define number generation as the correct visual depiction of an entity specified in a prompt and its associated quantity (e.g.,"2 red cats." or "One mushroom and 3 koalas."). To probe how well models capture *The abstraction principle*, we vary the context of numerical terms appear in. Then, we investigate whether and how image generation accuracy changes depending on the prompt structure (e.g., attributes, and compositionality) enumerated below to give seven different prompt types listed together with examples in rows "Exact" in Table 1.

**Prompt structure:** The simplest setting consists of phrases with an object and a number (*numeric-simple*). We explore different prompt structure of sentences (*numeric-sentence*), and also consider prompts which contain two or three number-noun combinations (*2-additive* and *3-additive*). We then include color adjectives for both simple prompts with one entity (*attribute-color*), and two entities with associated colors (*2-additive-color*). Finally, we combine number terms with spatial relationships such as "There are three cats above one manatee." (*attribute-spatial*).

**Exact number generation:** For the simplest setting (*numeric-simple*), we additionally examine the role of three other factors: (i) *Number magnitude* for which we generate text prompts with numbers ranging from 1 to 10. (ii) *Number representation* where we consider prompts that represent digits with both Arabic numerals (e.g., 1, 2, 3) or words (e.g., "one", "two", "three"). (iii) *Noun frequency* in English for which we select nouns (i.e., entities) to cover both frequent and rare words.[2]

---

[2]We use the wordfreq Python library [28] to determine word frequency. Our vocabulary in this task consists of approx. 40 words from four common categories: food, animals, nature, and objects. See Appendix A for more details.

Table 1: Twelve prompt types, example prompts and templates to probe different aspects of numerical reasoning in text-to-image models.

| Task | Prompt Type | Prompt Template / Example Prompts |
|---|---|---|
| Exact | numeric-simple | <num> <noun>
*One pistachio.*
*7 kangaroos.* |
| | numeric-sentence | There <verb> <num> <noun> in this image.
*There is 1 fish.*
*An image showing 1 fish.* |
| | 2-additive | <num1> <noun1> and <num2> <noun2>.
*3 durians and three bonsais.*
*Four axolotls and three cookies.* |
| | 2-additive-color | <num1> <color> <noun1> and <num2> <color> <noun2>.
*Four red cats and four red mushrooms.*
*One black cat and 1 black apple.* |
| | 3-additive | <num1> <noun1>, <num2> <noun2> and <num3> <noun3>.
*2 parsnips, 3 coconuts and one seahorse.*
*Two manatees, 2 burgers and 1 okra.* |
| | attribute-color | <num> <color> <noun>.
*One black koala.*
*4 green bottles.* |
| | attribute-spatial | There <verb> <num1> <noun1> <rel> <num2> <noun2>.
*There are three cats above one manatee.*
*There are three coconuts to the left of 5 burgers.* |
| Approx. | approx-1-entity | An image of <noun>. There <verb> <quant> <loc>.
*An image of a shelf. There are many books on the shelf.*
*An image of a watermelon. There are no seeds in the watermelon.* |
| | approx-2-entity | An image with some <noun1> and <noun2>. There <verb> <quant> <noun1> <quant> <noun2>.
*An image with some tables and some pencils. There are fewer tables than pencils.*
*An image with some bottles and some apples. There are as many bottles as apples.* |
| Quantitative | fractional-simple | There <verb> <num> <noun> and <frac> of another <noun>.
*A pizza cut into 3 slices.*
*There are three apples and quarter of another apple on the table.* |
| | fractional-complex | A <noun> is cut into <num> pieces. One piece is twice the size of the other.
*An image of a pencil where one half of it is red and the other half is blue.*
*A banana is cut into two pieces. One piece is twice the size of the other.* |
| | part-whole | There <verb> <num> <noun>, but one <noun> is broken into two pieces.
*There are 2 forks on the table, but one fork is broken into two pieces.*
*There are 5 pencils on the table, but one pencil is broken into two pieces.* |

## 3.2 Task 2: Approximate Number Generation and Zero

In Task 2, we evaluate models on their ability to correctly depict entities with quantities expressed in *approximate* terms by means of linguistic quantifiers (e.g., "many", "a few", or "more"). Such quantity terms are ubiquitous in ordinary language use and may denote a range of numbers thus carrying a more fuzzy interpretation of quantities. However, people tend to be relatively consistent when using such terms, as shown in Figure A16. We expect that models that correctly depict approximate quantities align with human perception of these quantities. We also examine how well models interpret the concept of zero, which we evaluate separately as it represents a challenging milestone in number learning in children [32].

**Annotation Task 1**    **Annotation Task 2**    **Annotation Task 3**

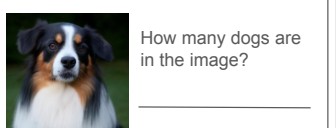 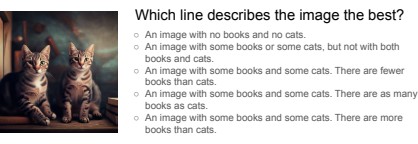 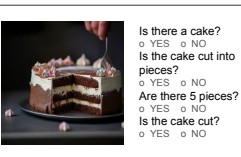

Figure 2: Three types of annotation templates used to collect data for the evaluation of text-to-image models on three numerical reasoning tasks.

**Prompt types.** We design two prompt templates for this purpose: *approx-1-entity* and *approx-2-entity*, as shown in Table 1. The former tests for approximate generation of one entity in the prompt (e.g., a few *candles*); the latter includes two entities with quantities expressed in relation to each other (e.g., more *books* than *cats*), as existing studies indicate poor performance on this prompt type [18].

### 3.3   Task 3: Conceptual Quantitative Reasoning

In this task we evaluate models on prompts that require a conceptual understanding of objects and their parts and are thus more challenging compared to the previous tasks. Notions of "parts", "proportion" and "fractions" tend to be concepts that are harder to acquire for both children and animals [26, 34]. For example, when a fork is broken into two parts, three- to four-year-old children count each discrete physical object as a separate fork [26].

**Prompt types.** Three prompt types are used in this task (the row "Quantitative" in Table 1): the *fractional-simple* category includes notions of a one whole, and basic fractions such as ½ and ¼ (e.g., "A cake cut into quarters."); the *fractional-complex* prompts include notion of fractions in relation to another attribute such as color or size (e.g., "one piece is twice the size of the other"). Finally, inspired by the fork experiment with children [26], we include *part-whole* prompt type where an object such as a fork, pencil or a plate is split into pieces.

## 4   Human Annotations of Images

We evaluate text-to-image models based on annotations collected from study participants who completed three different tasks, where each task corresponds to one of the three numerical reasoning tasks discussed in Section 3. An example template used for each annotation task is shown in Figure 2.

To evaluate models on **Task 1**, we ask participants to count the number of objects in a generated image. Specifically, we pair each generated image with an automatically generated question `How many <obj> are in the image?`, where `<obj>` refers to the noun in the source prompt used to generate the image. We ask one question for each noun in the prompt. Participants responded with a number representing the count of objects entered in free-form text format; where relevant, they were instructed to provide number ranges (e.g. 1–2, 2.5) or "10+" labels (see Appendix C.1 for detailed annotation instructions).

For **Task 2**, an image is paired with 3–5 lines of text describing the image. The number of lines depends on the original prompt, for *approx-1-entity* there were three lines, and for *approx-2-entity* there were 5 lines. Participants were asked to select the line that describes the image the best. The suggested lines were derived from the text prompt used to generate the image (see Figure 2 for an example).

Finally, to evaluate models on **Task 3**, we generate a series of short questions covering the words in the prompt words using automatic question generation approach based on Davidsonian Scene Graph (DSG) [9]. The questions appeared next to the image. DSG generates prompt-specific questions so that the answer to all of them should be "yes" if the model accurately depicted contents of the prompt. In analyses, we excluded all numerically irrelevant questions (i.e. `Is this an image?`, `Is there a table?`, `Are/is <obj> on the table?`). There are between two and six generated questions per each prompt.

We do not show the original prompt (used to generate a given image) to participants in any of the tasks, as we want to collect unbiased estimates of generated quantities. Example screenshots showing

the web interface participants were seeing for each annotation task are available in Figures A6-A8. Twenty-five participants have been recruited through a crowd-sourcing platform and provided informed consent to participate in the study. Our institution's independent ethical review committee reviewed and approved details of the study design, including working conditions and minimum hourly compensation of £15. Each image was annotated by five participants, and we observe a high level of inter-annotator agreement as in over 96% of cases at least 3/5 annotators provided the same rating (see Appendix C.2 for details).

**Processing human annotations.** Number annotations for object counts in Task 1 are first pre-processed by removing typos and standardizing the format (see Appendix C.2 for details on text preprocessing). The mode of the five numbers (i.e. the most frequent number) for each image–question pair is considered as the numerical "label" (i.e. model response) used to calculate model accuracy. When calculating accuracy, the ground-truth number is the original number in the text prompt used to generate the image. If the two numbers match, the accuracy is "1", otherwise it is "0".

In Task 2 each original prompt template is encoded as a number (e.g., "no <X>" would be encoded as "0", "many <X>" as "4" etc., see Appendix A.3) taken to be the ground-truth number. Each participant response, a radio-button selection, is also encoded as a number on the same scale. If the two numbers match, the accuracy is "1", otherwise it is "0".

In Task 3 the "yes"/"no" responses given by participants are coded as binary values 1/0 representing accuracy on the question. We aggregate all binary responses and to obtain the average accuracy for a given prompt–image pair.

# 5 Evaluating Text-to-Image Models

**Text-to-image models.** We study 12 different text-to-image models across five different model families: DALL·E 3 [2], Midjourney, Imagen-based models [25, 31], Muse [4] and Stable Diffusion (SD) models [21, 23, 24]. The models we evaluate cover a wide span of model architectures, including pixel-based (e.g. Imagen) and latent-based (e.g. Muse, SD) models. Specifically, the Muse family uses a different generative approach based on predicting masked image patches.

Models within each family may differ in their size, architecture and training data. For models that have been trained on internal data sources we use letters of the alphabet to refer to earlier versions of models (i.e. Imagen-A is an earlier version of Imagen-B, and Imagen-D is a recent version referred to as Vermeer [31]). Generally, the core components of earlier Imagen models have fewer parameters compared to later models (e.g. 600M for Imagen-A, 2B for Imagen-B and Imagen-C, and 8B for Imagen-D). Imagen A, B and C models have been trained on internally curated versions of WebLI dataset [7], while Imagen-D has been trained on CC12M [5]. In all our experiments, for each model we generate five images using five different seeds.

Table 2: Per task accuracy and the standard error of the mean, with percentage point difference from the baseline in brackets. Accuracy of the best performing model on a task is highlighted in bold, and that of the best model within a family is underlined.

| | **Task 1** Exact Number Generation | **Task 2** Approximate Number Generation and Zero | **Task 3** Conceptual Quantitative Reasoning |
|---|---|---|---|
| DALL·E 3 | **45.2** $\pm$ 0.5 $(+35.2\%)$ | **48.7** $\pm$ 2.7 $(+24.1\%)$ | 48.8 $\pm$ 1.1 $(-1.2\%)$ |
| Midjourney v6 | 42.5 $\pm$ 0.5 $(+32.5\%)$ | 35.0 $\pm$ 2.3 $(+10.4\%)$ | 48.9 $\pm$ 1.1 $(-1.1\%)$ |
| Imagen-A | 26.3 $\pm$ 0.4 $(+16.3\%)$ | 20.0 $\pm$ 2.2 $(-4.6\%)$ | 41.1 $\pm$ 1.3 $(-8.9\%)$ |
| Imagen-B | 27.0 $\pm$ 0.4 $(+17.0\%)$ | 24.6 $\pm$ 2.3 $(+0.0\%)$ | 42.9 $\pm$ 1.4 $(-7.1\%)$ |
| Imagen-C | $\underline{34.9}$ $\pm$ 0.4 $(+24.9\%)$ | 27.0 $\pm$ 2.4 $(+2.4\%)$ | $\underline{\mathbf{50.6}}$ $\pm$ 1.2 $(+0.6\%)$ |
| Imagen-D | 28.5 $\pm$ 0.4 $(+18.5\%)$ | $\underline{28.7}$ $\pm$ 2.4 $(+4.0\%)$ | 43.8 $\pm$ 1.3 $(-6.2\%)$ |
| Muse-A | 34.8 $\pm$ 0.4 $(+24.8\%)$ | 21.0 $\pm$ 2.2 $(-3.6\%)$ | 45.1 $\pm$ 1.2 $(-4.9\%)$ |
| Muse-B | $\underline{39.8}$ $\pm$ 0.5 $(+29.8\%)$ | $\underline{24.6}$ $\pm$ 2.3 $(+0.0\%)$ | $\underline{46.2}$ $\pm$ 1.2 $(-3.8\%)$ |
| SD1.5 | 20.3 $\pm$ 0.4 $(+10.3\%)$ | 20.6 $\pm$ 2.2 $(-4.0\%)$ | 44.6 $\pm$ 1.2 $(-5.4\%)$ |
| SD2.1 | 25.8 $\pm$ 0.4 $(+15.8\%)$ | 27.9 $\pm$ 2.4 $(+3.3\%)$ | 43.5 $\pm$ 1.1 $(-6.5\%)$ |
| SDXL | 22.8 $\pm$ 0.4 $(+12.8\%)$ | 31.2 $\pm$ 2.5 $(+6.6\%)$ | 43.8 $\pm$ 1.1 $(-6.2\%)$ |
| SD3 | $\underline{40.0}$ $\pm$ 0.5 $(+30.0\%)$ | $\underline{33.9}$ $\pm$ 2.6 $(+9.3\%)$ | $\underline{47.8}$ $\pm$ 1.0 $(-2.2\%)$ |
| Random Chance | 10.0 | 24.6 | 50.0 |

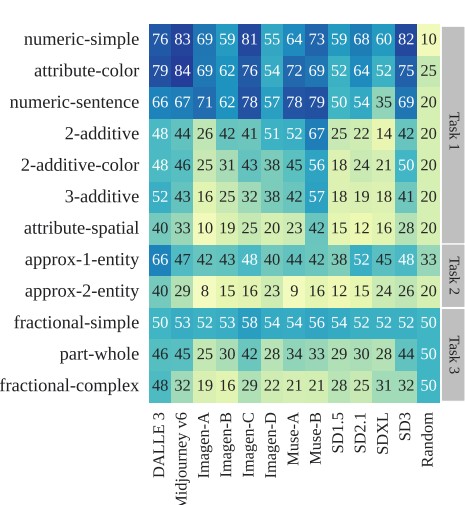

Figure 3: Accuracy of models on each prompt type for a subset of prompts that contain small numbers (i.e. 1–4) and a smaller subset of nouns.

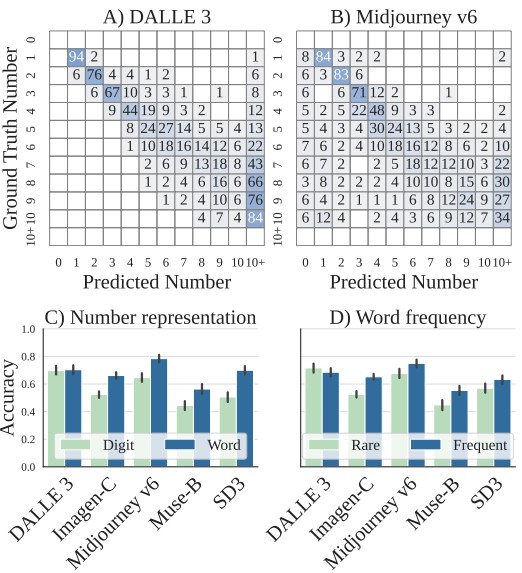

Figure 4: Top: The confusion matrices for A) DALL·E 3 and B) Midjourney v6 on *numeric-simple* prompts. Bottom: The effect of C) number representation and D) word frequencies in Task 1. 95% bootstrap confidence intervals are shown.

## 5.1 Task 1: Exact Number Generation

**Number magnitude.** We find that number magnitude strongly affects model performance in number generation, consistent with existing work in the multimodal and generative models [22, 20]. Figure 4 shows normalized confusion matrices for DALL·E 3 and Midjourney v6, the two strongest models on this task. Even for DALL·E 3, the accuracy drops substantially with each successive number (18 p.p. decrease for $1 \rightarrow 2$, 9 p.p. for $2 \rightarrow 3$, and 23 p.p. for $3 \rightarrow 4$). All models have a tendency to overestimate numbers (i.e. they tend to depict a higher number of entities than what is specified in the prompt), as shown by the non-zero entries over the "10+" label on the x-axis in the figure. With the exception of DALL·E 3, we also see underestimation, including instances where models fail entirely to generate the entity in the prompt (non-zero entries over the "0" label).

**Number representation (digits vs. words).** Based on *numeric-simple* prompts and the small number range where models have higher accuracy (1–4), we find that 10 out of 12 models are significantly more accurate when numbers in text prompts are represented with words as opposed to digits, with the exception of DALL·E 3 and SD1.5, where there was no significant difference. Figure 4C) shows accuracy for a subset of best performing models in each family for Task 1.

**Noun frequency (frequent vs. rare words).** Nine out of 12 models were more accurate when the nouns associated with numbers in text prompts were frequent words, as opposed to rare words, for *numeric-simple* prompts. The only exceptions were DALL·E 3, Imagen-D and SDXL, where there was no significant difference in accuracy. While this finding may indicate that these three models are better at abstracting the notion of a number as opposed to memorizing frequently occurring number-noun combinations, we note that this finding is based on a small set of words ($N = 40$) that may or may not generalize for a greater sample size. The comparison of accuracies for a subset of models is shown in Figure 4Ḋ).

**Prompt structure.** Compositional prompts types with more than one number-noun combination, such as *2-additive* and *3-additive*, were on average more difficult compared to prompts with a single noun and a number (i.e. *numeric-simple*) (c.f. Figure 3). With the exception of DALL·E 3, performance of all other eleven models was also significantly lower when numbers appeared in the same context with **spatial relationships** as in *attribute-spatial* prompts, as those prompts were significantly harder when compared to *2-additive* prompts that did not include spatial relationships. Adding **color** terms to numbers significantly reduced accuracy in Imagen-C, SD1.5 and SD3 models. These results indicate that prompt complexity, such as including additional prepositions or numbers in the text, can dramatically impair the ability of models to correctly generate images containing even a small numbers of entities. In Section B.2 in the Appendix we discuss different types of model failures in Task 1, and in Section C.3.1 we discuss the occasional cases of disagreements in human annotations with several qualitative examples.

## 5.2 Task 2: Approximate Number Generation and Zero

When testing for approximate number generation, similarly to Task 1, we observe that prompts with fewer entities (i.e. *approx-1-entity*) are on average easier than prompts with two entities (i.e. *approx-2-entity*), as seen in Figure 3. Figure 5 shows a further breakdown of accuracy per specific prompt template. For *approx-1-entity* (first 3 rows containing only "X"), Eleven out of 12 models were the most accurate in generating "many" objects, and all models were least accurate in generating images with zero objects (i.e. "There is/are no X"). For *approx-2-entity* (last 3 rows, with both "X" and "Y"), 8 out of 12 models have

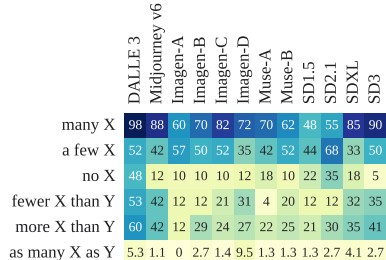

| | DALL·E 3 | Midjourney v6 | Imagen-A | Imagen-B | Imagen-C | Imagen-D | Muse-A | Muse-B | SD1.5 | SD2.1 | SDXL | SD3 |
|---|---|---|---|---|---|---|---|---|---|---|---|---|
| many X | 98 | 88 | 60 | 70 | 82 | 72 | 70 | 62 | 48 | 55 | 85 | 90 |
| a few X | 52 | 42 | 57 | 50 | 52 | 35 | 42 | 52 | 44 | 68 | 33 | 50 |
| no X | 48 | 12 | 10 | 10 | 10 | 12 | 18 | 10 | 22 | 35 | 18 | 5 |
| fewer X than Y | 53 | 42 | 12 | 12 | 21 | 31 | 4 | 20 | 12 | 12 | 32 | 35 |
| more X than Y | 60 | 42 | 12 | 29 | 24 | 27 | 22 | 25 | 21 | 30 | 35 | 41 |
| as many X as Y | 5.3 | 1.1 | 0 | 2.7 | 1.4 | 9.5 | 1.3 | 1.3 | 1.3 | 2.7 | 4.1 | 2.7 |

Figure 5: Accuracy for *approx-1-entity* and *approx-2-entity* prompts.

highest accuracy when generating images from prompts of the "more X than Y"-type and lowest accuracy when generating "as many X as Y". Our results highlight that understanding of linguistic quantifiers, even as simple as the word "no", can still be challenging for generative models. Section C.3.2 in the Appendix shows qualitative examples including images and annotations in Task 2.

## 5.3 Task 3: Conceptual Quantitative Reasoning

Task 3 is the hardest, as most models perform close to or below the random chance baseline (c.f. Table 2). All twelve models had the highest accuracy on *fractional-simple* prompts, followed by *part-whole* and *fractional-complex* for 10 out of 12 models, consistent with intuitions regarding the prompt difficulty. The last column in Figure 1 shows images for a prompt in the *fractional-simple* category, where many models failed to correctly depict "one apple and half of another apple". Further examples, including images and annotations, are shown in Table A15.

However, evaluating models on Task 3 prompts was more challenging due to the question-based method used—for some prompts, the DSG method generates only two relevant questions out of which one asks about the presence of an object in the image (e.g. *Are there apples?*) and thus answering "yes" yields the baseline of 50%, without necessarily capturing the nuance between whole apples, sliced apples or thirds/quarters of an apple. Further methodological challenges related to evaluation of models on this task are discussed in Appendix C.4.

## 6 Measuring What Counts: Challenges in Evaluation of Numerical Reasoning

We explore the utility of GECKONUM for two related areas of research, developing automatic evaluation metrics and evaluating vision–language models (VLMs) with respect to numerical reasoning

**Evaluating auto-eval metrics.** Developing metrics that can reliably replace human evaluation is an active area of research. In particular, a recent line of work proposes *auto-eval metrics* for measuring text-to-image alignment by using pretrained language and/or VLMs [14, 15, 33, 9, 35]. Here, we evaluate such auto-eval metrics on their ability to capture

Table 3: The ability of auto-metrics to discriminate between correct and incorrect image generations. (✓ indicates $p < 0.05$ using Mann-Whitney U test)

|  | CLIP | TIFA | Gecko | DSG | VNLI |
|---|---|---|---|---|---|
| DALL·E 3 | = | = | ✓ | ✓ | ✓ |
| Midjourney v6 | = | ✓ | ✓ | ✓ | ✓ |
| Imagen-C | = | ✓ | ✓ | ✓ | ✓ |
| Muse-B | ✓ | ✓ | ✓ | ✓ | ✓ |
| SD3 | = | ✓ | ✓ | ✓ | ✓ |

exact number generation by examining how well they distinguish between correctly and incorrectly generated images for *numeric-simple* prompts containing small numbers (i.e. 1–4) in Task 1.

We consider five different auto-eval metrics: CLIPscore [14], a metric based on computing similarity between the text and an image, question-answering (QA) based metrics such as TIFA [15], Gecko [33], DSG [9], as well as VNLI [35], a metric fine-tuned to predict alignment between images and text. We divide images generated by DALL·E 3, Midjourney v6, Imagen-C, Muse-B, and SD3 into two groups each based on whether the image was generated correctly or not. Then, we statistically test whether the distributions of scores in the two groups is identical. If there is a statistically significant difference between the two distributions, and if the correct generations have a higher score, we mark that case with a ✓ in Table 3. Gecko, DSG and VNLI are able to reliably distinguish between correct and incorrect image generations for all models. Upon manual inspection of generated questions for each QA method, we observed that Gecko and DSG generate similar questions, while TIFA occasionally generates questions about concepts that do not exist in the prompt. Following the approach in [33], we use a Wilcoxon signed-rank test to test whether there is a significant difference in average scores between pairs of models for the same metric. Only VNLI and Gecko were able to correctly order models in pairwise comparisons, in 8/10 and 7/10 cases, respectively. We also note that this analysis is based on simple prompts which trigger simple questions and thus most of the heavy lifting in performance of an auto-metric is delegated to the underlying VQA model.

**Evaluating counting in vision-language models (VLMs).** While counting is known to be challenging for VLMs, only a few datasets and benchmarks exist to train and evaluate VLMs on counting [1, 29]. We explore if GECKONUM, which VLMs have never been explicitly trained on, can be used as an evaluation task by curating a VQA benchmark. We evaluate the accuracy of PaLIGemma [3], a state-of-the-art open-source VLM, on counting. Technical details about the setup and more detailed results of experiments in this section are available in Appendix D.

Comparing the base PaLIGemma model to a checkpoint that has been fine-tuned for counting on TallyQA (train) [1], we find that the fine-tuned model performs better ($73.3_{\pm 0.4}\%$) on GECKONUM when compared to the base model ($68.4_{\pm 0.4}\%$). We observe that the base model already performs well on small numbers (up to 4), but fine-tuning on another counting dataset (i.e., TallyQA) improves

accuracy on higher counts ( $\geq 5$), a pattern we also observe with GECKONUM. We also briefly investigate the utility of GECKONUM as synthetic training data: we fine-tune base PaLIGemma on a mixture of TallyQA (train) and images from all Imagen models, and evaluate it on TallyQA (test) and Muse-B images (see Appendix D for setup details). We find that including Imagen data does not significantly change performance on TallyQA (test) but vastly improves performance on Muse-B (by more than 20 p.p. in some cases). This is still true when we test on a set of held-out Muse-B classes that we removed from the Imagen fine-tuning data. These preliminary results highlight that: 1) training VLMs with synthetic data can improve results on other benchmarks; 2) there is great need for more public high-quality datasets and benchmarks that evaluate counting and numerical reasoning.

## 7  Discussion and Conclusion

Most work on evaluation of text-to-image models uses generic prompts—manually written or harvested form the Web—as a starting point to generate images. For example, recent research has focused on designing comprehensive prompt sets that measure text-to-image alignment [33, 15, 9, 16, 30, 13].

In this work, we propose GECKONUM, a benchmark that specifically focuses on measuring numerical reasoning capacity through three tasks: (i) exact and (ii) approximate number generation, and (iii) conceptual reasoning about quantities. We define various prompt types within these tasks to better control for the effect of factors such as the context of the number in the sentence and the sentence structure. We use GECKONUM to evaluate twelve different text-to-image models from five different model families (DALL·E 3, Midjourney v6, Imagen, Muse and Stable Diffusion) by collecting human annotations for the generated images. We find that DALL·E 3 has the highest overall accuracy on exact and approximate number generation, while being the least impacted by some prompt manipulations we investigate (e.g. numbers represented as words vs. digits, frequent vs rare words); however, its performance on these tasks is still close to or under 50%. Depiction of approximate quantities and the concept of zero was also a weak point in all models. For example, even DALL·E 3 would consistently fail to correctly generate "a watermelon with no seeds" or "a cake with no candles". We also find that the task requiring reasoning about parts and fractions was challenging for all models as their performance was close to the baseline.

We used GECKONUM to show that only some auto-eval metrics can reliably differentiate between correct and incorrect images on simple numerical prompts—among metrics we tested, our preliminary experiments show that VNLI [35] and Gecko [33] are the only two metrics capable of such differentiation, including the ability to rank models in pairwise model comparisons. We also demonstrated that our benchmark could be used to study and potentially improve the performance of pretrained vision–language models on counting.

Our approach relies on human annotators to provide counts of objects in an image, as is often the gold standard in evaluation of text-to-image models. However, such annotation process is laborious, costly and does not scale. We expect that the rapid improvement of pipelines and frameworks for evaluation of text-image alignment will reduce the need for manual annotation [10, 35, 13]. In addition, we identified three important numerical capabilities for which we have manually designed human annotation templates, but numerical cognition in humans spans a wide set of gradually developing capabilities that might require additional evaluation templates. As discussed in Section 5.3, the design of protocols for evaluation of more complex aspects of numerical reasoning remains an important open challenge with rapid advancements in models.

Overall, our results indicate that the current text-to-image models do not form abstract representations of numbers as their ability to reason about numbers is rudimentary: it is limited to depiction of exact, small quantities in images, and models frequently fail to generate approximate quantities and zero. We highlight the effectiveness of numerical reasoning as an open challenge in evaluation, since even in the large-scale training regime, due to the large combinatorial space of numbers, it is challenging to mitigate models' weaknesses with more training data. Future modeling innovations—and not only better training data—might be needed to improve model performance on numerical reasoning. Our results show that most models are sensitive to the number representation in a given prompt (words vs digits). This showcases another interesting challenge involved in evaluating for numbers: they can occur in various formats (e.g., dates, phone numbers, or expressions such as "4k") and representations. Also, effective ways of tokenizing numbers, again due to the vast space of possible numbers, is an open problem in language modeling [27] and an interesting direction for future research.

**Acknowledgments.** We are grateful to our colleagues at Google DeepMind for advice and feedback that helped improve this work. We thank Cristina Vasconcelos for a detailed review of the manuscript, Jason Baldridge for support and useful feedback, Richard Tucker and Aayush Upadhyay for their help and technical support essential in realization of this work, and to Cyrus Rashtchian and Matko Bošnjak for valuable feedback.

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
