# Appendices

# A  Additional Information on Prompts

Table A1: Nouns used in text prompts in Task 1.

| Frequent words | Rare words |
|:---:|:---:|
| apple | lychee |
| egg | parsnip |
| burger | samosa |
| cookie | cinnamon stick |
| pizza | axolotl |
| dog | seahorse |
| fish | kangaroo |
| cat | koala |
| ant | manatee |
| fly | mushroom |
| tree | durian |
| leaf | bonsai |
| flower | pistachio |
| coconut | okra |
| olive | crib |
| table | paperclip |
| book | flute |
| bottle | trowel |
| spoon | corkscrew |
| pencil | |
| chocolate | |

## A.1  Words and Word Frequencies

Nouns used in text prompts in Task 1 are selected from the list of 40 words, listed in Table A1. The words cover different semantic categories such as everyday objects, food, nature and animals. Approximately the half of them ($N = 21$) are labelled as frequent, and the rest ($N = 19$) as non-frequent (i.e. rare), as word frequency was a variable we investigated in Section 5.1. The distribution of frequencies for those words is shown in Figure A1, and frequencies are obtained using the Python wordfreq library [28] that gathers frequency data based on multiple sources of data such as Wikipedia, OPUS OpenSubtitles 2018 [19] and Google Books Ngram, among many others. For the word 'cinnamon stick' we used the word 'cinnamon' which was present in the database. In Tasks 2 and 3 we mostly used the same words as in Task 1, but in Task 3 also added a few additional objects/foods such as "fork", "a loaf of bread" or "cake", where we investigated how well text-to-image models can generate parts of objects or fractions.

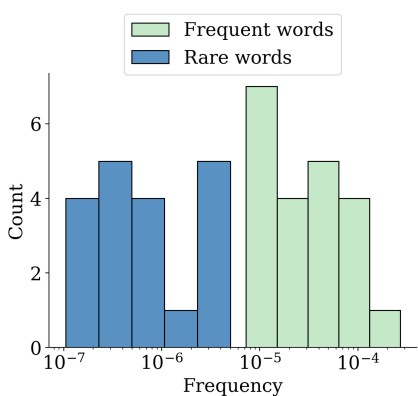

Figure A1: The distribution of frequencies for the nouns used in text prompts in Task 1.

## A.2 The Distribution of Prompt Types in the Benchmark

Table A2: The distribution of prompt types in the benchmark ($N = 1386$)

| Task | Prompt Type | Number of prompts | Numbers |
|---|---|---:|---|
| Task 1 | numeric-simple | 600 | 1, 2, 3, 4, 5, 6, 7, 8, 9, 10 |
| | attribute-color | 160 | 1, 2, 3, 4 |
| | numeric-sentence | 100 | 1, 2, 3, 4, 5 |
| | 2-additive | 100 | 1, 2, 3, 4, 5 |
| | 2-additive-color | 100 | 1, 2, 3, 4, 5, 6, 7, 8 |
| | 3-additive | 100 | 1, 2, 3, 4, 5 |
| | attribute-spatial | 100 | 1, 2, 3, 4, 5 |
| Task 2 | approx-1-entity | 24 | no, few, many |
| | approx-2-entity | 45 | fewer, as many as, more |
| Task 3 | fractional-simple | 36 | 1, 2, 3, 1/2, 1/3, 1/4, 1/5 |
| | part-whole | 15 | 1/2 |
| | fractional-complex | 6 | 1/3 + 2/3, 1/2 |

## A.3 The Encoding Scheme for Task 2 Answers

- 0: An image with no X and no Y. // There are no X.
- 1: An image with some X or some Y, but not with both X and Y.
- 2: There are fewer X than Y. // There are only a few X.
- 3: There are as many X as Y.
- 4: There are more X than Y. // There are many X.

# B Additional and Detailed Experimental Results

In this section we show results for all models for experiments in Section 5. In significance tests we use $\alpha = .05$ when rejecting the null hypothesis based on the p-value of a test. When Chi-squared test is used to compare accuracies between two groups representing two different conditions (i.e. "digits" vs "numerals", or "frequent" vs "rare" words), we build a contingency table based on binary accuracy counts for prompt–image pairs in different groups. In those cases, the null hypothesis is that there is no significant difference in accuracy between the two groups. We report accuracies as percentages in tables for easier comparison with other values in the manuscript. As a control condition, i.e. to confirm that the variable we investigate indeed explains the differences in results, we also conducted permutation tests where we randomly permuted labels between the two groups under comparison. Unless indicated differently in the text in those conditions we found that all differences in accuracies were not significant.

## B.1 Task 1: Exact Number Generation

### B.1.1 Number magnitude

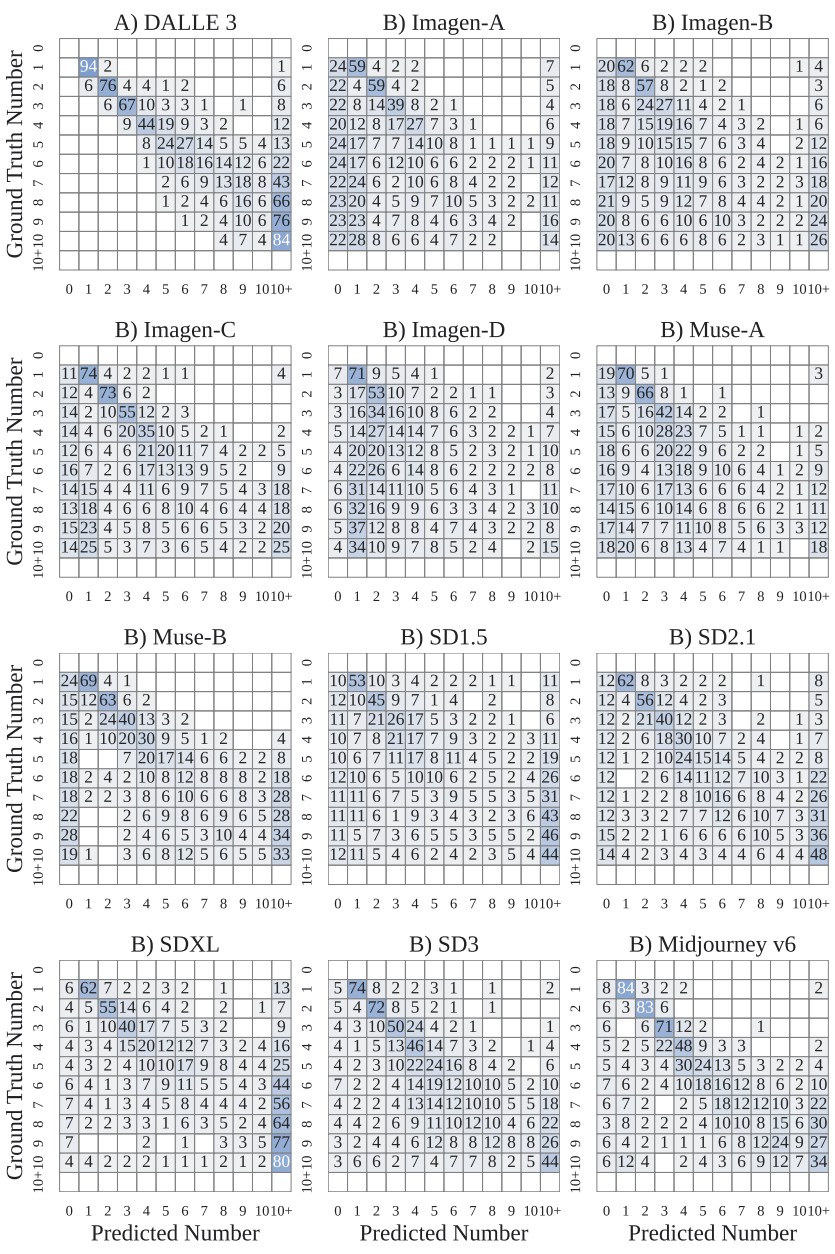

Figure A2: The confusion matrices for all models for *numeric-simple* prompt type. The generated number of objects in an image (as annotated by humans) is on the x-axis, and the ground truth number in the original text prompt is shown on the y-axis. Empty cells denote 0.

## B.1.2 Number representation and word frequency

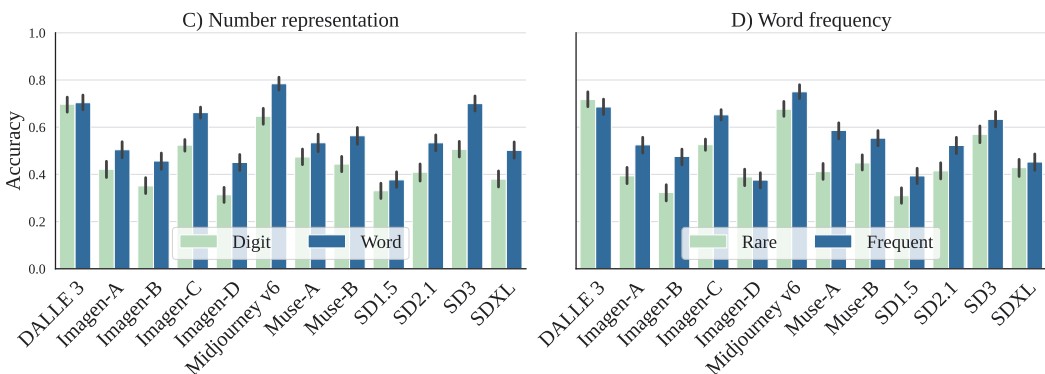

Figure A3: Number representation (A) and word frequency (B) accuracy for *numeric-simple* prompts and all models. Number range: 1–4. Lines on top of bars show 95% bootstrapped confidence intervals.

.

Table A3: Chi-squared test results for comparison of accuracy between prompts with digits ($Acc_D$) and prompts with word numerals ($Acc_W$). N: Number of samples, $\chi^2$: Chi-squared statistic value, Diff?: whether the difference between two accuracies is significant, ✓: significant difference, =: insignificant difference.

|  | p-value | $Acc_D$ | $Acc_W$ | N | $\chi^2$ | Diff? |
|---|---|---|---|---|---|---|
| DALLE 3 | 0.827 | 69.8 | 70.4 | 1600 | 0.05 | = |
| Midjourney v6 | 0.000 | 64.6 | 78.4 | 1592 | 36.42 | ✓ |
| Imagen-A | 0.001 | 42.2 | 50.4 | 1592 | 10.60 | ✓ |
| Imagen-B | 0.000 | 35.1 | 45.7 | 1595 | 17.95 | ✓ |
| Imagen-C | 0.000 | 52.4 | 66.2 | 3178 | 61.85 | ✓ |
| Imagen-D | 0.000 | 31.4 | 45.1 | 1596 | 31.09 | ✓ |
| Muse-A | 0.018 | 47.4 | 53.4 | 1594 | 5.55 | ✓ |
| Muse-B | 0.000 | 44.4 | 56.4 | 1600 | 22.56 | ✓ |
| SD1.5 | 0.064 | 33.1 | 37.7 | 1577 | 3.44 | = |
| SD2.1 | 0.000 | 40.9 | 53.4 | 1599 | 24.36 | ✓ |
| SD 3 | 0.000 | 50.6 | 70.0 | 1600 | 61.92 | ✓ |
| SDXL | 0.000 | 38.0 | 50.2 | 1564 | 23.09 | ✓ |

Table A4: Chi-squared test results for comparison of accuracy between prompts with rare words ($Acc_R$) and prompts with frequent words ($Acc_F$). N: Number of samples, $\chi^2$: Chi-squared statistic value, Diff?: whether the difference between two accuracies is significant, ✓: significant difference, =: insignificant difference.

|  | p-value | $Acc_R$ | $Acc_F$ | N | $\chi^2$ | Diff? |
|---|---|---|---|---|---|---|
| DALLE 3 | 0.189 | 71.7 | 68.6 | 1600 | 1.73 | = |
| Midjourney v6 | 0.001 | 67.6 | 75.0 | 1592 | 10.22 | ✓ |
| Imagen-A | 0.000 | 39.4 | 52.5 | 1592 | 26.94 | ✓ |
| Imagen-B | 0.000 | 32.3 | 47.6 | 1595 | 38.04 | ✓ |
| Imagen-C | 0.000 | 52.6 | 65.2 | 3178 | 51.90 | ✓ |
| Imagen-D | 0.621 | 38.9 | 37.6 | 1596 | 0.24 | = |
| Muse-A | 0.000 | 41.2 | 58.6 | 1594 | 47.71 | ✓ |
| Muse-B | 0.000 | 44.9 | 55.4 | 1600 | 17.14 | ✓ |
| SD1.5 | 0.001 | 31.0 | 39.4 | 1577 | 11.72 | ✓ |
| SD2.1 | 0.000 | 41.5 | 52.3 | 1599 | 18.10 | ✓ |
| SD3 | 0.011 | 57.0 | 63.3 | 1600 | 6.48 | ✓ |
| SDXL | 0.373 | 42.9 | 45.2 | 1564 | 0.79 | = |

### B.1.3 Prompt structure: Additive prompts

Table A5: Percentage point drop in accuracy for a specific number (1–3) present in different prompt types.

| | Number | SN→2A | 2A→3A | SN→3A |
|---|---|---|---|---|
| DALLE 3 | 1 | -9.7 | -3.2 | -12.9 |
| | 2 | -19.2 | -10.4 | -29.6 |
| | 3 | -14.2 | -9.5 | -23.7 |
| Imagen-C | 1 | -17.4 | -5.6 | -23.0 |
| | 2 | -35.4 | -0.1 | -35.5 |
| | 3 | -21.8 | -9.5 | -31.4 |
| Midjourney v6 | 1 | -27.9 | 9.3 | -18.6 |
| | 2 | -16.7 | -13.0 | -29.7 |
| | 3 | -17.9 | -10.8 | -28.6 |
| Muse-B | 1 | -0.1 | -10.3 | -10.4 |
| | 2 | -20.7 | 4.2 | -16.5 |
| | 3 | -1.6 | -4.1 | -5.7 |
| SD3 | 1 | -4.7 | -1.7 | -6.4 |
| | 2 | -32.8 | -5.0 | -37.8 |
| | 3 | -18.5 | -6.0 | -24.6 |

Figure A4 shows drops in accuracy when a specific number occurs in different prompt types. In addition, Table A5 shows the exact drops in accuracy for selected models for the numbers: "1", "2" and "3". While for some models the drop in accuracy for the number "1" (c.f. the three bars above the x-tick "1") is much smaller (e.g. DALL·E 3) or non-significant (Muse-B) compared to accuracy drops for some other models (e.g. Imagen-A, Imagen-C), all models show a substantial and significant drop when "2" is present in the prompt (c.f. the three bars above the x-tick "2").

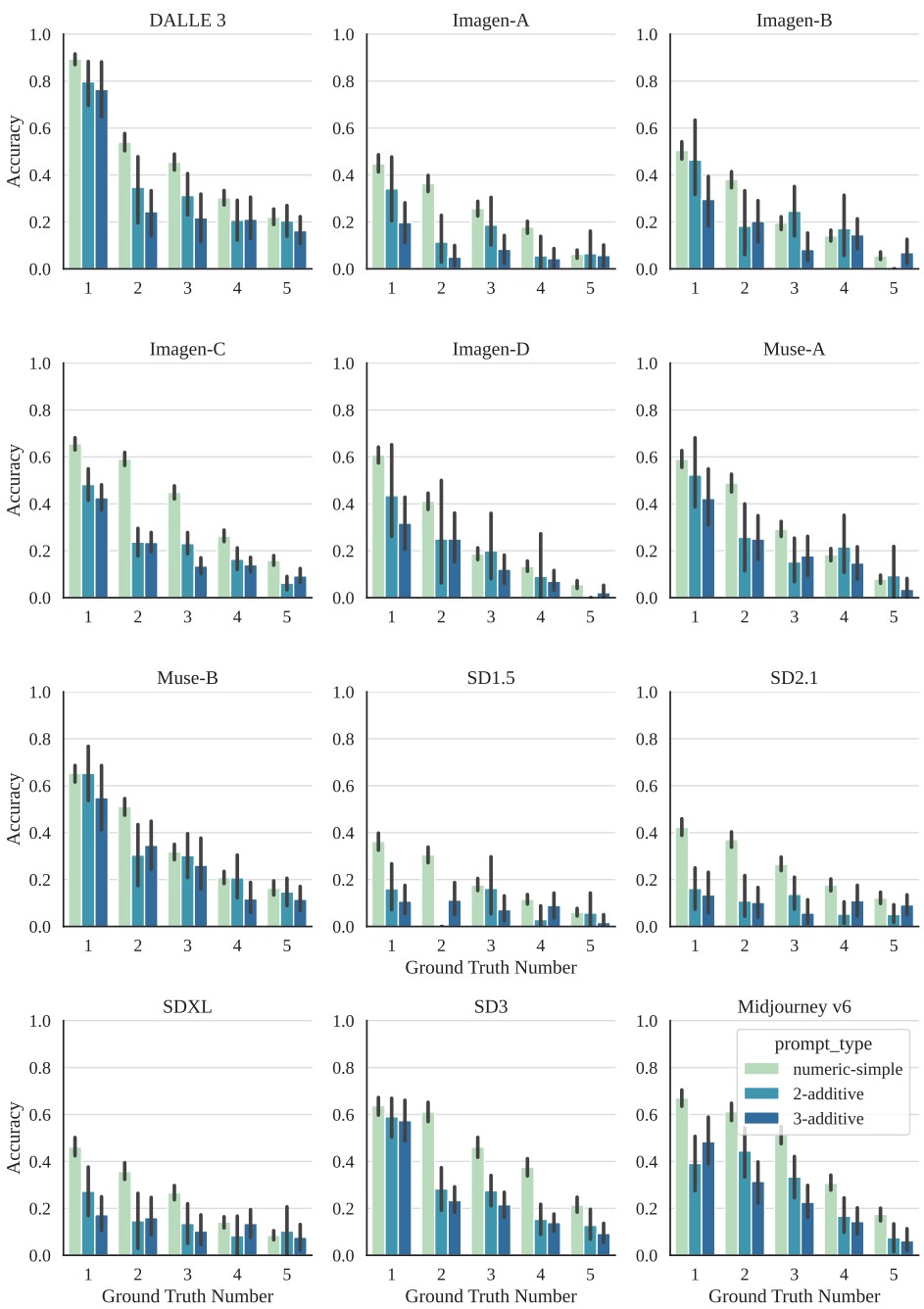

Figure A4: Accuracy per ground truth number in the prompt for different additive prompt types. Lines on top of bars show 95% bootstrapped confidence intervals.

.

### B.1.4 Prompt structure: Colors and spatial relationships

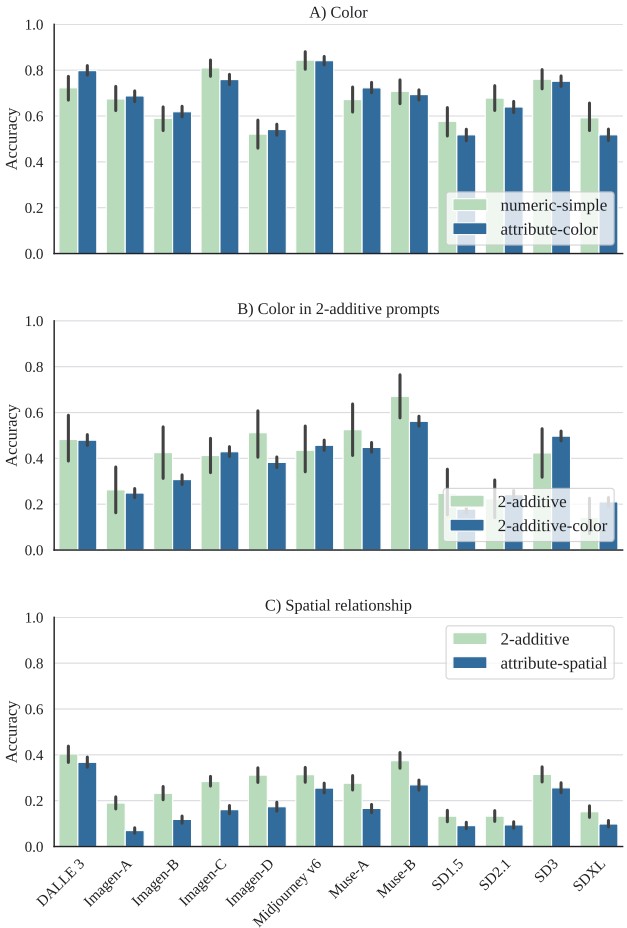

Figure A5: Accuracy in exact number generation when: A) color terms are added to number terms in the prompt, B) color terms are added in 2-additive prompts, and C) spatial relationships are introduced. Lines on top of bars show 95% bootstrapped confidence intervals.
.

To investigate how the addition of color terms and spatial relationships affects accuracy in number generation we ask the following three questions:

1. Does adding color adjectives to numbers (such as "1 red koala") affect the number generation accuracy?

2. Does adding color adjectives to numbers in 2-additive prompts (such as "1 red koala and two green cats") affect the number generation accuracy?

3. Does the introduction of spatial relationships between nouns associated with numbers affect the number generation accuracy?

To answer the first question, we split the data into the two groups: the first group contains the subset of data for *numeric-simple* prompts, and the second group the subset of data for *attribute-color* prompts. We only consider prompts in both groups that contain the same numbers (1–4) and the same words ("cat", "apple", "koala", "bottle", "mushroom"), to isolate the effect of adding the color term as opposed to potential confounding factors. For example a confounding factor might be the word identity, as a model might be more accurate in generating correct images when the prompt contains the word "dog", and if this word exists only in the first prompt type and not in the second then responses in the first prompt type will on average have higher accuracy that may or may not

depend on color terms. We split the data in a similar way for the other two questions. To answer the second question we split the data based on *2-additive* and *2-additive-color* prompts, and to answer the third question we split the data into *2-additive* and *attribute-spatial* prompts.

We show the average accuracy between groups for each of these conditions in Figure A5, and significance test results based on Chi-squared test in Tables A6-A8. We see that most differences were not significant when the color term was added, with the exception of Muse-A in Figure A5A) where the accuracy was actually higher when the color term was added, and Imagen-B and Imagen-D in Figure A5B). In both Figures A5A) and B) we highlight relatively large confidence intervals for *numeric-simple* and *2-additive* groups. Those two types of prompts were associated with fewer datapoints, compared to the groups they were compared to (i.e. *attribute-color* and *2-additive-color*, respectively). While the average accuracies are much closer in Figure A5A), we speculate that with more samples in Figure A5B) we may see more significant differences for some other models. In contrast, when the prompt contained a spatial relationship, we see significant differences for all models except for DALL·E 3 (Figure A5C).

Table A6: Chi-squared test results for comparison of accuracy between prompts with no color terms ($Acc_{NC}$, i.e. *numeric-simple*) and prompts with color terms ($Acc_C$, i.e. *attribute-color*). N: Number of samples, $\chi^2$: Chi-squared statistic value, Diff?: whether the difference between two accuracies is significant, ✓: significant difference, =: insignificant difference.

| | p-value | $Acc_{NC}$ | $Acc_C$ | N | $\chi^2$ | Diff? |
|---|---|---|---|---|---|---|
| DALLE 3 | 0.470 | 0.77 | 0.79 | 1800 | 0.52 | = |
| Midjourney v6 | 0.603 | 0.83 | 0.84 | 1796 | 0.27 | = |
| Imagen-A | 0.994 | 0.69 | 0.69 | 1785 | 0.00 | = |
| Imagen-B | 0.499 | 0.59 | 0.62 | 1800 | 0.46 | = |
| Imagen-C | 0.024 | 0.81 | 0.76 | 1999 | 5.07 | ✓ |
| Imagen-D | 0.717 | 0.55 | 0.54 | 1795 | 0.13 | = |
| Muse-A | 0.020 | 0.64 | 0.72 | 1797 | 5.37 | ✓ |
| Muse-B | 0.298 | 0.73 | 0.69 | 1800 | 1.08 | = |
| SD1.5 | 0.048 | 0.59 | 0.52 | 1775 | 3.91 | ✓ |
| SD2.1 | 0.389 | 0.68 | 0.64 | 1800 | 0.74 | = |
| SD3 | 0.040 | 0.81 | 0.75 | 1794 | 4.20 | ✓ |
| SDXL | 0.053 | 0.60 | 0.52 | 1787 | 3.73 | = |

Table A7: Chi-squared test results for comparison of accuracy between 2-additive prompts with no color terms ($Acc_{2A}$) and 2-additive prompts with color terms ($Acc_{2AC}$). N: Number of samples, $\chi^2$: Chi-squared statistic value, Diff?: whether the difference between two accuracies is significant, ✓: significant difference, =: insignificant difference.

| | p-value | $Acc_{2A}$ | $Acc_{2AC}$ | N | $\chi^2$ | Diff? |
|---|---|---|---|---|---|---|
| DALLE 3 | 1.000 | 0.48 | 0.48 | 2015 | 0.00 | = |
| Midjourney v6 | 0.778 | 0.44 | 0.46 | 2015 | 0.08 | = |
| Imagen-A | 0.883 | 0.26 | 0.25 | 2010 | 0.02 | = |
| Imagen-B | 0.036 | 0.42 | 0.31 | 2006 | 4.42 | ✓ |
| Imagen-C | 0.750 | 0.41 | 0.43 | 2086 | 0.10 | = |
| Imagen-D | 0.023 | 0.51 | 0.38 | 2014 | 5.16 | ✓ |
| Muse-A | 0.212 | 0.53 | 0.45 | 2006 | 1.56 | = |
| Muse-B | 0.061 | 0.67 | 0.56 | 2015 | 3.50 | = |
| SD1.5 | 0.144 | 0.25 | 0.18 | 2007 | 2.14 | = |
| SD2.1 | 0.804 | 0.22 | 0.24 | 2007 | 0.06 | = |
| SD3 | 0.225 | 0.42 | 0.50 | 2015 | 1.47 | = |
| SDXL | 0.177 | 0.14 | 0.21 | 2012 | 1.82 | = |

Table A8: Chi-squared test results for comparison of accuracy between prompts without spatial relationships ($Acc_{NS}$, i.e. *2-additive*) and prompts with spatial relationships ($Acc_S$ i.e. *attribute-spatial*). N: Number of samples, $\chi^2$: Chi-squared statistic value, Diff?: whether the difference between two accuracies is significant, ✓: significant difference, =: insignificant difference.

|  | p-value | $Acc_{NS}$ | $Acc_S$ | N | $\chi^2$ | Diff? |
|---|---|---|---|---|---|---|
| DALLE 3 | 0.095 | 0.40 | 0.37 | 2350 | 2.79 | = |
| Midjourney v6 | 0.003 | 0.31 | 0.25 | 2350 | 8.67 | ✓ |
| Imagen-A | 0.000 | 0.19 | 0.07 | 2313 | 74.82 | ✓ |
| Imagen-B | 0.000 | 0.23 | 0.12 | 2315 | 50.00 | ✓ |
| Imagen-C | 0.000 | 0.28 | 0.16 | 3084 | 66.50 | ✓ |
| Imagen-D | 0.000 | 0.31 | 0.17 | 2341 | 56.94 | ✓ |
| Muse-A | 0.000 | 0.28 | 0.17 | 2315 | 37.80 | ✓ |
| Muse-B | 0.000 | 0.37 | 0.27 | 2350 | 27.00 | ✓ |
| SD1.5 | 0.003 | 0.13 | 0.09 | 2325 | 8.94 | ✓ |
| SD2.1 | 0.006 | 0.13 | 0.09 | 2346 | 7.63 | ✓ |
| SD3 | 0.003 | 0.31 | 0.26 | 2348 | 8.74 | ✓ |
| SDXL | 0.000 | 0.15 | 0.10 | 2328 | 13.87 | ✓ |

## B.2 Qualitative Analysis of Model Failures

Table A9: Examples of images where models failed to generate one object as specified in the prompt. See the text for details.

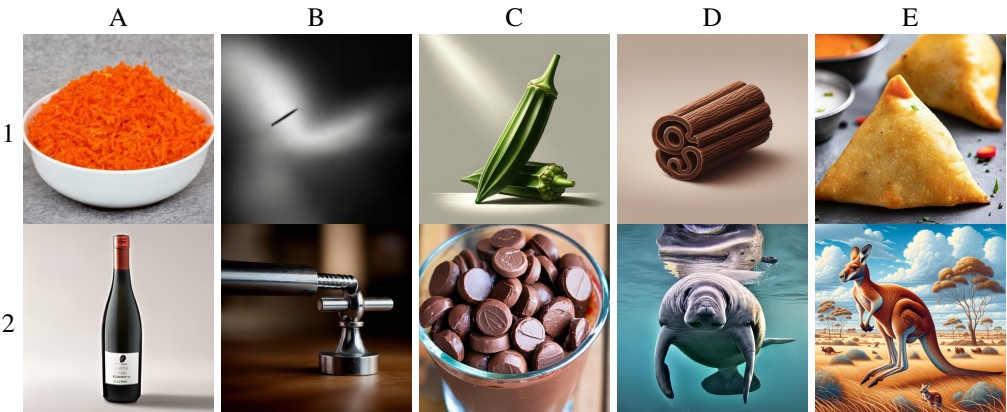

We manually inspected a subset of images for *numeric-simple* prompts with the ground truth "1" (i.e. the prompts of the format: `1 <noun>.`, or `One <noun>.`) to understand different types of model failures in this most basic case. One trivial kind of failure, especially observed with some earlier models such as Imagen-A and Muse-A, is not generating the object at all (column A in Table A9). For example, in A1) the model failed entirely to generate a *spoon*, while in A2) it failed to generate a *corkscrew*. In other instances, the generated object may have some characteristics of the object in the prompt, such as a silhouette of a *pencil* in B1), or the same material, such as metal of a *corkscrew* in B2).

In many other instances, we found that models generate more than one object, such as two *okras* in C1) or many *chocolates* in C2). Among models we evaluated, Imagen models were most susceptible to generating more than one object when that object was a pistachio, an olive or a cookie. There were also some more nuanced cases of model failures, subject to interpretation and details of our methodology. For example, for the *cinnamon stick* in D1), annotators counted 2, 2, 0, 1.5, 1.5, while for the *manatee* in D2) the counts were: 1.5, 1.5, 1, 2, 1. In both of these examples, rounding up the numbers and taking the most frequent response results in the number "2" as the label for that image, although the raw annotations reveal potential ambiguity in the image.

Finally, and as discussed in Appendix C.3, there were certain "edge cases" in counting such as objects that are shown in the background (column E) or shown only partially, where it is possible to count objects in different ways. For example, in E1), it seems that all annotators assumed that the object in the background is another *samosa*, while in E2) they counted all *kangaroos* that appeared in the image. A few cases similar to E2) were observed with DALL·E 3, where the model would generate one dominant entity in the foreground, and several other, much smaller ones in the background. While our instructions specify that such objects should be counted separately (see Appendix C.1 and Figure A9), but we found that sometimes they were not counted separately or that this was difficult to do as the boundary between foreground and background may not be apparent in all such images.

We also inspected failure cases in other prompt types. For *additive* prompts we found that DALL·E 3 and Muse-B often correctly depicted two or three entities as specified in the prompt while failing to depict the correct number of those entities, while some smaller models such as Muse-A and Imagen-A frequently omitted one of the entities entirely. For *attribute-spatial* prompts, which were generally one of the hardest prompts, we notice that DALL·E 3 frequently generates a correct (or close to correct) number of entities, but not necessarily in the required spatial arrangement, in contrast to some models that fail to generate those objects at all.

### B.3 Technical Resources

To generate images for all Imagen models and Muse-A, we used a cluster system containing NVIDIA L4 Tensor Core GPUs. For Muse-B, and training experiments in Section 6 we used internal hardware accelerators. We estimate that we used 140 GPU hours to generate all images.

## C Collecting Human Annotations

### C.1 Annotation Instructions

All human annotators received training before completing the annotation tasks. They were instructed on how to use the web interface, were given task-specific instructions and were shown several examples of expected annotations with explanations. After the training, we conducted a pilot study with a small subset of images to confirm that task instructions were understood and followed. All images shown to annotators were passed through safety filters, and flagged images were manually inspected to ensure that no offensive, harmful or otherwise problematic content would be present in images. We did not collect any personally identifiable information.

**Annotation Task 1 Instructions.** Annotation instructions, including a few examples of edge cases in counting are shown in Figure A9. The goal of including a few examples of edge cases was to establish a guideline on how to count objects in cases where several interpretations are possible. For example, if it is possible that two shown halves of an object come from the same object we instructed annotators to count it as one object. If it is unlikely that they come from the same object, we instructed them to count the two parts as two different objects. Here, our aim was to provide guidance to reasonably standardize responses in such cases, but we did not aim to exhaustively cover all possible cases as we also wanted annotations to reflect individual differences in object perception in such less precise or ambiguous cases. If there were more than 10 objects present in the image, the instruction was to respond with "10+". In this task, responses were provided as free-form text. An example screenshot from this task is shown in Figure A6.

**Annotation Task 2 Instructions.** Instructions shown to annotators in the second task are shown in Figure A10. Annotators were told to provide a response quickly while being accurate, to encourage them to estimate quantities on this task, as opposed to counting each object in the image separately. We wanted to get a response that best reflects judgement of approximate quantities, as opposed to exact quantities that were the subject of interest in Task 1. An example screenshot from this task is shown in Figure A7.

**Annotation Task 3 Instructions.** Figure A11 shows instructions and an example annotation in Task 3, as well as an additional example screenshot in Figure A8.

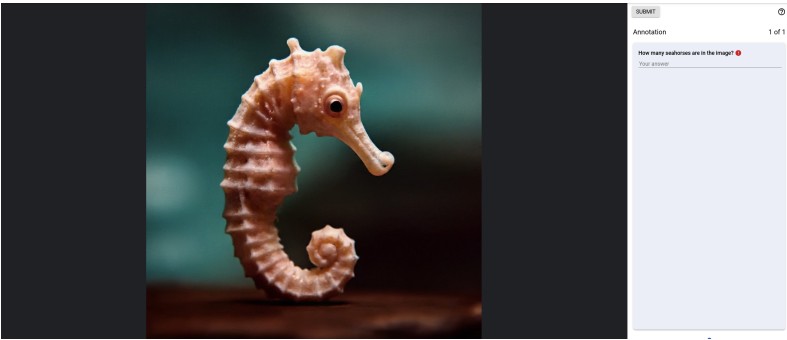

Figure A6: A screenshot of the user interface for one annotation example for Annotation Task 1. The response to the question is given in a free-form text format.

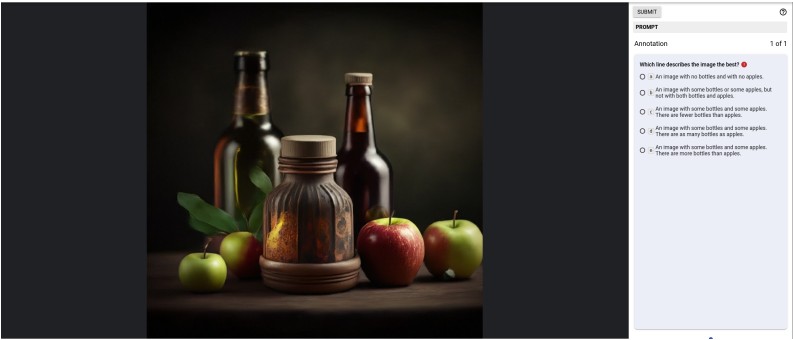

Figure A7: A screenshot of the user interface for one annotation example for Annotation Task 2. The response to the question is a single radio-button choice.

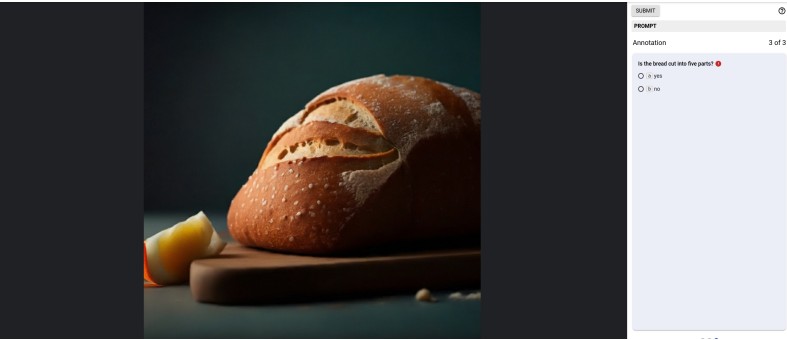

Figure A8: A screenshot of the user interface for one annotation example for Annotation Task 3. The response to the question is a single radio-button choice with yes/no options. In this example, there were three questions for this image, and this was the third question.

## Instructions (1/2)

- Count the number of objects asked in the question and type the number in the text form

- In most cases the answer will be a **single number**
  - If needed, use a decimal point to indicate a half of an object (eg 2.5)

- In some cases, it may be difficult to give an exact number, respond with a **range of possible values** (eg. 2-5), or **10+** if there are more than 10 objects

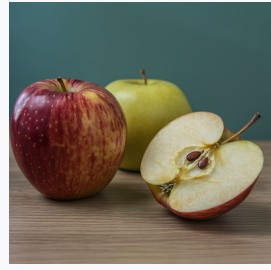

**How many apples are in the image?**

2.5

## Instructions (2/2)

- Sometimes, we may clearly see the same type of object being located in **foreground** and **background**

- If that is the case, the response should be two numbers separated by a comma and a space (eg. 2, 5)
  - One number for the number of objects in the foreground
  - One number for the number objects in the background

- Prefer to give one number when it is not clear which objects are in the foreground and which are in the background

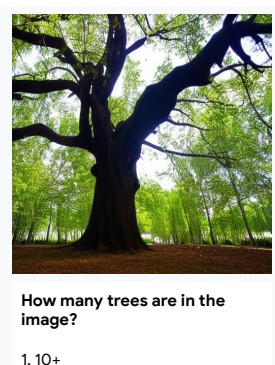

**How many trees are in the image?**

1, 10+

**Explanation of this response:**
There is 1 tree is clearly visible in the foreground, and many (10+) trees in the background.

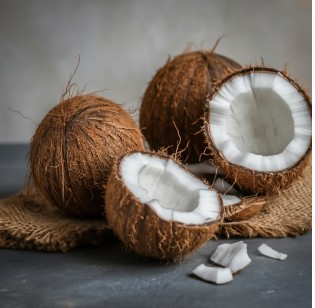

**How many coconuts are in the image?**

3-4

**Explanation of this response:**
If we count that one coconut is split in half then "3" makes sense, but if we count that two halves come from a different coconut then 4 makes sense, so we give a range of possible values: 3-4

Figure A9: Annotation instructions including examples of edge cases in counting in Task 1.

## Instructions

- Read all the lines and select the one line that describes image the best

- Try to provide a response as quickly as you can while being accurate
  - Use your best judgement to estimate the quantities of objects in the image without counting each object individually

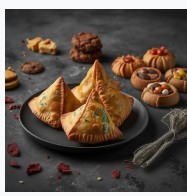

**Which line describes the image the best?**

- An image with no samosas and with no cookies.
- An image with some samosas or some cookies, but not with both samosas and cookies.
- An image with some samosas and cookies. There are as many samosas as cookies.
- An image with some samosas and cookies. There are fewer samosas than cookies.
- An image with some samosas and cookies. There are more samosas than cookies.

20

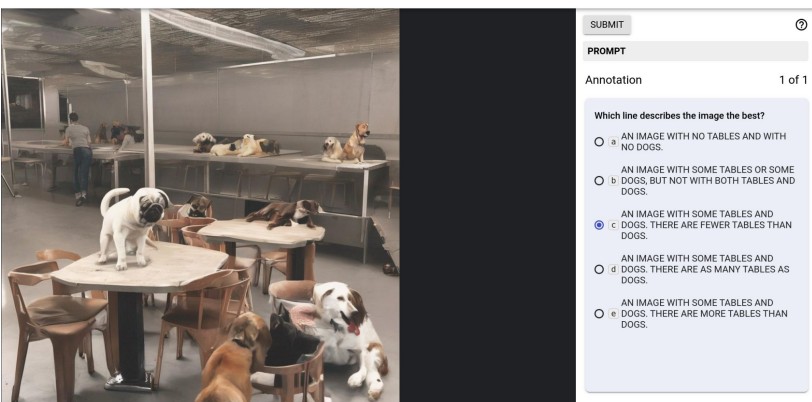

21

Figure A10: Annotation instructions including an example Task 2.

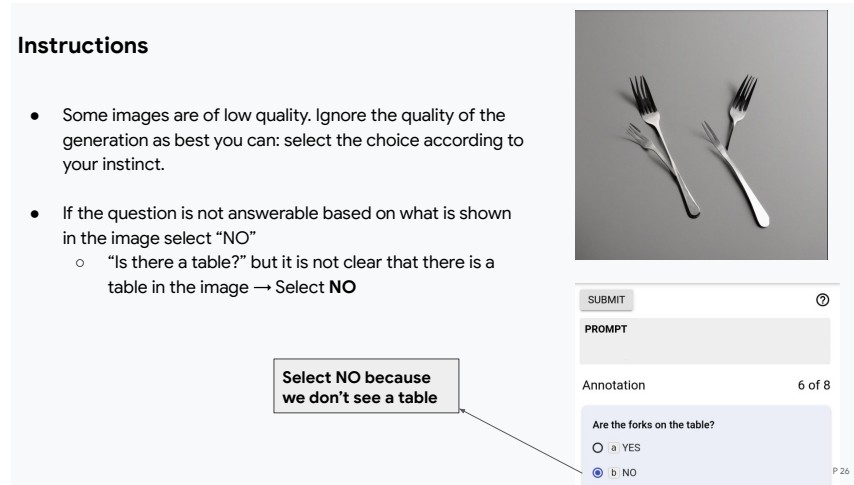

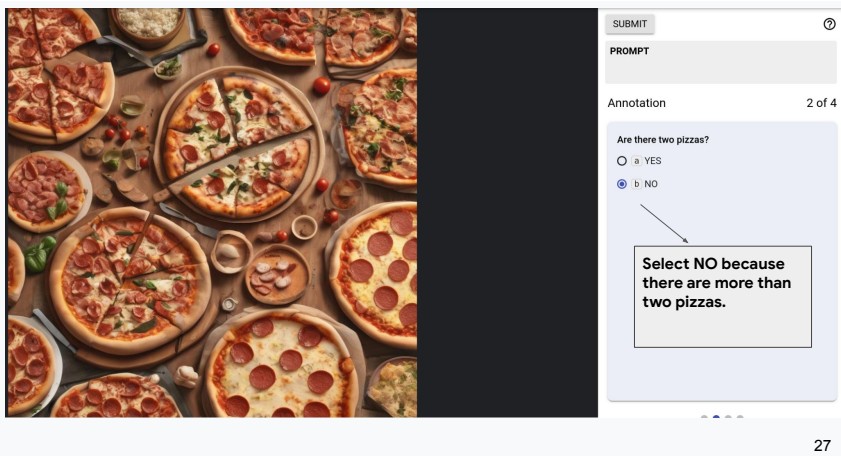

Figure A11: Annotation instructions including examples in Task 3.

## C.2 Quantitative Analysis

Table A10: Percentage of annotations where n out of 5 annotators provided the same response in Annotation Task 1.

|  | 1/5 | 2/5 | 3/5 | 4/5 | 5/5 |
|---|---|---|---|---|---|
| DALLE 3 | 0.0 | 2.3 | 6.5 | 8.5 | 82.7 |
| Midjourney v6 | 0.1 | 2.4 | 6.6 | 9.3 | 81.7 |
| Imagen-A | 0.1 | 2.3 | 6.6 | 8.7 | 82.3 |
| Imagen-B | 0.1 | 2.7 | 7.4 | 9.4 | 80.4 |
| Imagen-C | 0.1 | 2.8 | 8.1 | 10.4 | 78.7 |
| Imagen-D | 0.1 | 3.8 | 7.6 | 9.4 | 79.1 |
| Muse-A | 0.2 | 3.2 | 8.6 | 11.0 | 77.0 |
| Muse-B | 0.1 | 3.3 | 10.1 | 12.0 | 74.5 |
| SD1.5 | 0.2 | 4.2 | 10.1 | 11.5 | 73.9 |
| SD2.1 | 0.1 | 4.5 | 11.8 | 12.6 | 71.0 |
| SDXL | 0.3 | 5.1 | 9.7 | 11.0 | 73.9 |
| SD3 | 0.2 | 3.8 | 9.7 | 11.3 | 75.0 |

Figure A12: Distribution of numbers in ground-truth prompts used to generate images, and numbers given as counts of objects in annotated images.

In total, in Annotation Task 1 we collected 718,729 annotations, ranging from approximately $56-58$K annotations per model, except for Imagen-C where we collected 91K annotations as five additional images per prompt were generated for evaluation and fine-tuning experiments in Section 6 (for a total of 10 seeds for Imagen-C). In Annotations Tasks 2 and 3, we collected 20,960 and 62,010 annotations, respectively. For Imagen-C, the analyses and statistics below will be based on using five seeds for each model, although the data we release also includes additional images and annotations. Released data will not contain any images showing faces (approximately 122 images), due to potential privacy concerns, but analyses below were done including annotations collected for those images.

**Response processing.** From raw responses we removed spaces and all non-numeric characters, except for the letters "o"/"O" which we replaced with the number 0, "10+" entries were replaced 11 to get a numerical format, and ranges (i.e. 2–4) were converted into an integer by rounding up the average of the two values. When two comma-separated numbers were given, we took only the first number. If after these steps the response was an empty string, we removed it from the dataset. In total, only 10 such responses were removed.

**Response aggregation.** Each image and each question was shown to five annotators, collecting five annotations in total per image–question pair. In Task 1, *additive* prompts had several questions, one for each entity in the image, while in Task 3 there were several questions derived from the original text prompt. To get a single number representing the number of objects in an image we used the mode of five numbers in Tasks 1 and 2. In Task 2 those numbers were numerically encoded single-choice radio button options. In total, after aggregating responses there were 143,748 labels in Task 1, and 4192 labels in Task 2. In Task 3, we averaged all responses across all questions to get a score and refer to it as "accuracy" for consistency in analyses. Figure A12 shows normalized distributions of numbers in the original text prompts (green) and the distribution of numbers in annotated data (blue).

**Annotator agreement.** We observe high level of agreement among annotators. In Task 1, for each image–question pair, all 5/5 raters gave the same response in 77.5% of cases, at least 4/5 raters gave the same response in 87.9% of cases and at least 3/5 in 96.5%. In Task 2, the analogous percentages were as follows: 74.7%, 89.5% and 99.3%, and in Task 3 where we only had binary responses all 5/5 raters agreed in 84.9% of cases, while at least 4/5 agreed in 94.9%. Expressed as Krippendorff's alpha coefficient of inter-annotator agreement, $\alpha = 0.865$ for Task1, $\alpha = 0.863$ for Task 2, and $\alpha = 0.903$ for Task 3. A breakdown per model annotator agreements in Task 1 is shown in Table A10.

| Example # | Prompt | Image | Questions | Answers |
|-----------|--------|-------|-----------|---------|
| 1 | 10 coconuts. |  | How many coconuts are in the image? | 3.5, 3.5, 4, 4, 3.5 |
| 2 | 5 cinnamon sticks. |  | How many cinnamon sticks are in the image? | 0, 0, 5.5, 0, 6 |
| 3 | 5 trees. |  | How many trees are in the image? | 7.5, 7.5, 8, 8.5, 11 |
| 4 | 8 paperclips. |  | How many paperclips are in the image? | 8, 6.5, 9, 7, 6 |
| 5 | Two ants. |  | How many ants are in the image? | 1.5, 2, 1.5, 2, 2.5 |
| 6 | 9 koalas. |  | How many koalas are in the image? | 3.5, 4, 3.5, 2.5, 4 |
| 7 | 8 eggs. |  | How many eggs are in the image? | 4, 0, 4, 4, 0 |
| 8 | Four white apples. |  | How many apples are in the image? | 3.5, 4, 3.5, 4, 3-4 |
| | | | How many white apples are in the image? | 3.5, 0, 0, 0, 3.5 |
| 9 | There are 4 chocolates to the left of one fish. |  | How many chocolates are in the image? | 6, 6, 6, 6, 6 |
| | | | How many fish are in the image? | 1, 1, 1, 1, 1 |
| | | | How many chocolates are to the left of fish? | 4, 0, 0, 2, 0 |
| | | | How many fish are to the right of chocolates? | 1, 1, 1, 1, 1 |
| 10 | 1 leaf. |  | How many leaves are in the image? | 7, 7, 7, 7, 1 |

Figure A13: Examples of images and annotations in Annotation Task 1. Only images and questions were shown to annotators when completing the task.

| Example # | Prompt | Image | Answer (Best line match) |
|---|---|---|---|
| 1 | An image with some trowels and some manatees. There are as many trowels as manatees. |  | some trowels or some manatees, but not with both
no trowels and no manatees
as many trowels as manatees
some trowels or some manatees, but not with both
more trowels than manatees |
| 2 | An image with some fish and some seahorses. There are more fish than seahorses. |  | more fish than seahorses
fewer fish than seahorses
as many fish as seahorses
some fish or some seahorses, but not with both
as many fish as seahorses |
| 3 | An image with some kangaroos and some cribs. There are as many kangaroos as cribs. |  | some kangaroos or some cribs, but not with both
some kangaroos or some cribs, but not with both
more kangaroos than cribs
some kangaroos or some cribs, but not with both
some kangaroos or some cribs, but not with both |
| 4 | An image of a vase. There are no flowers in the vase. |  | many flowers in the vase
many flowers in the vase
no flowers in the vase
only a few flowers in the vase
only a few flowers in the vase |
| 5 | An image with some eggs and some burgers. There are as many eggs as burgers. |  | some eggs or some burgers, but not with both
some eggs or some burgers, but not with both
some eggs or some burgers, but not with both
as many eggs as burgers
some eggs or some burgers, but not with both |
| 6 | An image of a watermelon. There are many seeds in the watermelon. |  | no seeds in the watermelon
many seeds in the watermelon
no seeds in the watermelon
no seeds in the watermelon
no seeds in the watermelon |

Figure A14: Examples of images and annotations in Annotation Task 2. Only images, the question "Which line describes the image the best?" and 3–5 lines of text were shown to annotators when completing the task. The exact answers annotators selected are shortened for brevity.

| Example # | Prompt | Image | Questions and Answers |
|---|---|---|---|
| 1 | There is 1 plate on the table, but it is broken into two pieces. |  | Is there a plate? 1, 1, 1, 1, 1

Is there one plate? 0, 0, 0, 1, 1

Are there pieces of a plate? 1, 1, 1, 0, 0

Are there two pieces of a plate? 0, 0, 0, 0, 0

Is the plate broken? 1, 1, 1, 0, 0 |
| 2 | There are 2 pencils on the table, but one pencil is broken into two pieces. |  | Are there pencils? 1, 1, 1, 1, 1

Are there 2 pencils? 0, 0, 0, 0, 0

Is one pencil broken? 0, 1, 1, 0, 0

Is the pencil broken into 2 pieces? 0, 0, 0, 0, 0 |
| 3 | A loaf of bread cut into thirds. |  | Is there a loaf of bread? 1, 1, 1, 1, 1

Is the loaf of bread cut into thirds? 0, 1, 0, 0, 1 |
| 4 | An apple cut into 5 pieces. |  | Is there an apple? 1, 1, 1, 1, 1

Is the apple cut into pieces? 1, 1, 1, 1, 1

Are there 5 apple pieces? 1, 1, 1, 1, 1

Are the pieces inside the apple? 0, 1, 1, 1, 0 |
| 5 | An image of a pencil where one half of it is red and the other half is blue. |  | Is there a pencil? 1, 1, 1, 1, 1

Is there half of a pencil? 0, 0, 0, 0, 0

Is the pencil in two halves? 0, 0, 0, 0, 0

Is one half of the pencil red? 0, 0, 1, 1, 0

Is the other half of the pencil blue? 0, 0, 1, 1, 0 |
| 6 | An image of a pencil where one half of it is red and the other half is blue. |  | Is there a pencil? 1, 1, 1, 1, 1

Is there half of a pencil? 0, 0, 0, 0, 0

Is the pencil in two halves? 0, 0, 0, 0, 0

Is one half of the pencil red? 1, 1, 1, 1, 1

Is the other half of the pencil blue? 1, 1, 1, 1, 1 |

Figure A15: Examples of images and annotations in Annotation Task 3. Only images and questions were shown to annotators when completing the task. The answers are encoded as numbers: "1" denotes "yes", and "0" denotes "no".

## C.3 Qualitative Analysis

While the overall level of disagreement among annotators was low, as shown in Table A10, we manually inspected annotations to understand potential reasons for annotator disagreement and ensure the overall quality of the annotations. Examples for each task are shown in Figures A13–A15, and below we discuss specific cases and the overall observed trends.

### C.3.1 Task 1: Exact Number Generation

Figure A13 shows 10 selected examples of annotations with a varying level of disagreement in Task 1. We selected these examples as a representative subset of images, and discuss some common reasons why we believe there was some level of disagreement related to the perception of objects or quantities in an image. Some of these observations, such as that most people only count objects when those objects are clearly identifiable in an image, also held for the analysis of disagreement in Tasks 2 and 3.

First, we find that some disagreement may arise from a different interpretation of instructions, such as two annotators counting one half of a coconut as a whole, while three annotators counting it as a half, as per instructions (Example 1 in Table A13). In other cases, where there is some ambiguity regarding the identity of objects, it appears that annotators only counted objects when they were certain that shown objects matched the identity of objects mentioned in the question (Example 2 in the table). There were images where it was difficult to count objects based on how salient such objects were in an image, such as the trees shown in Example 3. We attempted to account for some such cases in instructions (such as dividing objects between those in the foreground and those in the background), but even then the boundary between foreground and background was not always apparent in generated images. Sometimes objects appeared distorted (Example 4), or there were morphed objects in an image and it was not possible to distinguish between individual entities (Examples 5 and 6).

There was a noticeable discrepancy in labels for images containing eggs, such as in Example 7. Specifically, we noticed that all annotators counted eggs when those appeared in shells or they were boiled, but if eggs were cracked (in the form of an egg whites and yolk) or fried, they would frequently not be counted as eggs. When we asked our annotators about the reasons for counting eggs in this way, we were told that some annotators perceived fried eggs as a dish, distinct from the concept of a "raw egg".

We also found that some level of disagreement stems from individual differences in color perception, such as "white apples" in Example 8. Finally, we also found that there were rare instances of incorrect annotations. For example in Example 9 we expect the correct answer to be "4" for the third question "How many chocolates are to the left of the fish?", but only one response matched our expectation. As well, Example 10 shows another instance where there is some ambiguity in interpretation, as the expected answer here is "1" and only one response matched the expectation.

### C.3.2 Task 2: Approximate Number Generation and Zero

Figure A14 shows six examples of selected images and corresponding annotations where there was some level of disagreement among annotators. We highlight these examples as representative of the types of disagreements we see in the data. When objects in images are not clearly identifiable people tend to indicate that one of the two objects is not present, such as Examples 1–3 in the figure.

In Examples 1 and 2, the model generated objects that were morphed forms of the two entities in the prompt. In Example 1, many of the objects appear to be approximately shaped like trowels, with one in the lower left corner being shaped like a manatee, but all objects have the color of a manatee. In this example, 3/5 people indicated that they do not recognize manatees, trowels or either. Somewhat similar trend is observed in Example 2, where generated objects appear to be a morphed combination of the two entities mentioned in the prompt–fish and seahorses. However, in this image more people were able to recognize objects, though from the data we have it is unclear how ambiguous objects were perceived.

In Example 3 we see another instance where the majority of annotators did not perceive an object when that object was not clearly identifiable. Specifically we see guard rails, but it is not clear that those rails belong to a crib, and so only one annotator indicated that there are more kangaroos, while the remaining four indicated that there are "no kangaroos or no cribs".

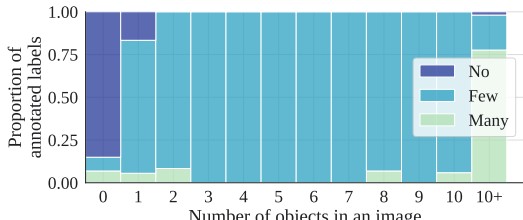

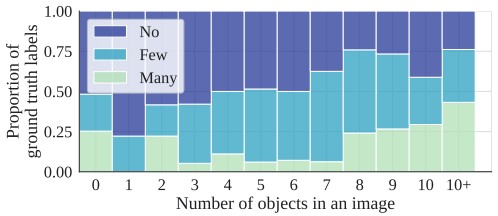

Figure A16: The proportion of "no", "few" and "many" annotated labels associated with a specific count of objects in an image.

Figure A17: The proportion of "no", "few" and "many" labels from the ground truth prompts associated with a specific count of objects in an image.

In Example 4, we see some individual differences in perception of quantity, where some people perceive the quantity of flowers as "many" while other as "only a few". In Example 5, and as in images in Task 1, we again observe that 4/5 annotators did not consider "fried eggs" as eggs, with the exception of one annotator. Finally, Example 6 is interesting because it highlights that people naturally expect to see watermelon seeds in a watermelon, even though the objects in the watermelon do appear as some kind of seeds.

As in this task we used linguistic quantifiers to study approximate quantities, there is some inherent subjectivity involved when interpreting the quantities. Specifically, as seen in Example 4, some people perceive that there are "many" flowers in the vase, while others perceive "only a few" flowers.

To further understand these individual differences, we collected additional annotations for images generated for *approx-1-entity* prompts in Task 2. These are the prompts that only contain one entity and the quantity associated with the entity is either "no", "[only a] few" or "many". Additional annotations were counts of objects in those images, as done in Task 1. Then, we paired annotations collected in Task 2, namely labels such as "no", "few" and "many" extracted from the lines annotators selected when completing the task with those counts, and plot the proportion of labels for each count in Figure A16.[3]

Despite differences in individual interpretations of linguistic quantifiers, we find that annotators were highly consistent in their interpretations of such quantities–the overwhelming majority of them selected "no" when they counted zero objects in an image (the dark blue bar over "0" in Fig. A16), and similarly, when there are more than 10 objects annotators labelled those images as "many". The notion of "few" had a more distributed span ranging from the numbers 1 to 10, but for counts in the range 3–7 100% of labels were "few".

We performed a similar analysis for labels extracted from the ground truth prompts, instead of annotations, for a comparison on how generated quantities in images correspond to labels. The results are shown in Figure A17. We indeed see a trend where the bars for "no" are more skewed towards the left, meaning more models generated fewer items when the word "no" was in the prompt, and analogously we see more "many" items toward the right.

### C.3.3 Task 3: Conceptual Quantitative Reasoning

Figure A15 shows six examples of images and corresponding annotations in Task 3. We generally see some ambiguity in responses when images are supposed to depict broken objects. For instance, in Example 1, 2/5 annotators responded that there is one plate in the image, while the other three annotators responded that there are pieces of a plate and that the plate was broken. Similarly in Example 2, 2/5 annotators perceive that a pencil is broken when pencil lead was cracked.

Some disagreement was present when talking about parts, as seen in Examples 3–5. While the loaf of bread in Example 3 is cut into three parts, the parts are not equal, but 2/5 annotators responded that the loaf of bread is cut into thirds. Examples 4 and 5 highlight the difficulty of model evaluation in this task. In Example 4, the question where annotators gave different responses ("Are the pieces inside the apple?") is neither meaningful nor relevant for evaluation of numerical reasoning, and some level of disagreement in this case might be expected.

---

[3]For each image we had five annotations, but we take the most frequent annotation as the label.

In Example 5, we speculate that there were individual differences in perception of "half of a certain color"—2/5 annotators did indicate that one half of the pencil is red and the other half is blue, while 3/5 did not. We manually inspected images and annotations for this specific prompt to see what would an image look like if all annotators agreed. One such example is shown in the last row of the table. We speculate that in answering this question, the annotators might have considered the color of pencil lead as well as the wood when answering the question.

Based on these and other similar examples, we conclude that disagreement in this task may come from ambiguity present in images, questions or both. The ambiguity in images may arise because objects and their parts are not clearly shown, as observed in Tasks 1 and 2, but also because it may not be clear whether objects are broken/sliced etc. The ambiguity in questions may arise because questions are not sufficiently specific, or in some cases, not meaningful.

## C.4 Methodological Challenges in Model Evaluation

In Tasks 1 and 2 we could directly express model accuracy by comparing whether the annotated count (e.g. "3") or quantity (e.g. "many") matched the corresponding count or quantity in the ground truth prompt used to generate the image. This was more difficult in Task 3, where prompts included objects that were shown in parts or pieces, and these parts or pieces might be associated with different properties, such as different sizes or colors. Text-to-image models which generate images that better depict more of such properties should be given a higher score, compared to those that match only some. For example, let us consider the following text prompt: *A pizza cut into 3 slices.*

Given this prompt, if a model A produces an image of a pizza that is not sliced at all, and a model B produces an image with a pizza cut into quarters, we would expect that model B is better aligned with the prompt as it correctly captured the notion of "cut" and "slice". For this reason, we decided to ask several questions that are grounded in the text prompt for each image in evaluation of models on this task. To obtain such questions, we use a recent automatic method based on the Davidsonian Scene Graph (DSG) [9]. This method generates questions based on words in the prompt in such a way that the expected answer to the question is "yes". For the example prompt above, such questions could be: *Is there a pizza?*, *Is the pizza cut?* and *Is the pizza cut into 3 slices?* Model B would score higher as it generated an image that can be answered with "yes" in 2/3 questions, while for model A this would only be the case for 1/3 questions.

While the majority of automatically generated questions are relevant in the context of evaluation of numerical reasoning, there are some limitations to this approach, as hinted in Section 5.3 and in Appendix C.3.3. First, some questions might be less discriminative for more capable text-to-image models that are able to clearly depict the object in the prompt. Specifically, it may be the case that for all models the answer to the question *Is there a pizza?* will be "yes". Second, occasionally the answer to some questions might be "no", as seen in Examples 5 and 6 in Table A15 where questions imply that the pencil might be split into halves (i.e. *Is there half of a pencil?* and *Is the pencil in two halves?*). We also noticed that some questions were not informative or were confusing, such as *Are the pieces inside the apple?* (Example 4 in Table A15) or *Is the quarter of a loaf of bread on the loaf of bread?* Based on manual inspection of the data, such questions accounted for a small proportion of all questions and we expect that as automatic methods advance their number will be even smaller in the future.

## D    VQA experiments

### D.1    The VQA Setup

In our VQA experiments we use PaLIGemma [3]. PaLIGemma is an open 3B vision-language model (VLM) inspired by PaLI-3 [8], built with open components, such as the SigLIP [37] vision model and the Gemma-2B language model [12]. We use the public checkpoint for an input resolution of $448 \times 448$ pixels as counting requires fine-grained visual information available at `https://www.kaggle.com/models/google/paligemma/jax/paligemma-3b-pt-448`.

To turn GECKONUM into a VQA dataset/benchmark, we added questions of the form `How many <noun>s are in the image?` to each data point. As the ground truth answer we used the mode of the answers provided by the annotators, as we did in the rest of the work.

For the evaluation benchmark we only included high-quality images on which at least four out of the five annotators gave the same response; in this case, the mode corresponds to the majority vote. This yields $59,582$ high-agreement question-answer pairs from Imagen-A+B+C+D and Muse-A+B. For this set of experiments only, we also generated additional images with Imagen-C, as it was one of the best performing models in this family. In total, we had 10 different images for each prompt for Imagen-C. For the fine-tuning experiments we included all images even when three or more annotators disagreed in their responses. We only trained on Imagen-A+B+C+D data, which corresponds to $53,053$ "noisy" question-answer pairs. In experiments where we split the dataset into "frequent" and "rare" classes, we use the filter that filters images based on the division of prompts as described in Appendix A.

To fine-tune PaLIGemma we found that fine-tuning for two epochs with a learning-rate of $10^{-5}$, a weight-decay strength of $10^{-6}$, and a batch-size of $256$ worked well across different training mixtures (otherwise we used the default hyper parameter settings). Due to the exploratory nature of these experiments, we did not re-tune hyper parameters for each experiment and setting separately, but we used a single setting for all fine-tuning experiments.

## D.2 Additional Results

### D.2.1 GECKONUM as a VQA benchmark

Table A11: Accuracy (in %) when evaluating PaLIGemma checkpoints on TallyQA (test) and on the GECKONUM VQA benchmark.

| PaLIGemma checkpoint | TallyQA test simple | TallyQA test complex | GECKONUM |
|---|---|---|---|
| Base | 35.4 | 37.8 | 68.4 |
| Fine-tuned (TallyQA train) | 84.9 | 72.1 | 73.3 |

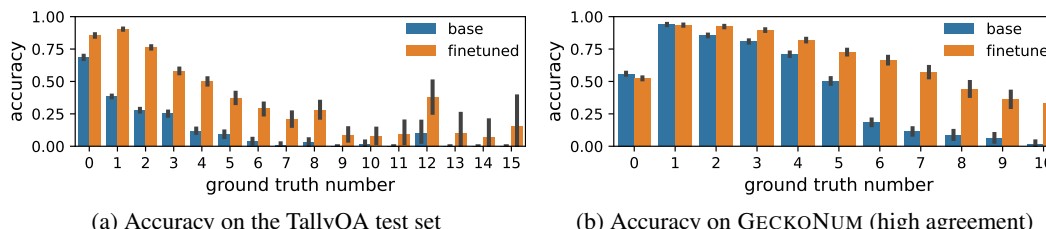

(a) Accuracy on the TallyQA test set     (b) Accuracy on GECKONUM (high agreement)

Figure A18: Accuracy of PaLIGemma (base model or fine-tuned on TallyQA (train)) when evaluated on the TallyQA test set (left) and GECKONUM (right) by ground truth number.

Here we first provide additional results on using GECKONUM as a VQA benchmark. We evaluate the base model and a model fine-tuned on TallyQA (train) on both TallyQA (test) as well as on GECKONUM, see Table A11 and Figure A18 for results.

We observe that GECKONUM seems to be an easier benchmark as the base model is already able to answer many questions correctly, especially for lower counts. This is likely because while TallyQA includes complex and cluttered scenes, GECKONUM images focus on one or very few object classes.

Moreover, we find that the fine-tuned model particularly improves on higher counts ($\geq 5$), as shown in Figure A18. We observe the same trend for both TallyQA (test) and GECKONUM; this indicates that fine-tuning on TallyQA transfers to other benchmarks and datasets and that PaLIGemma fine-tuned has indeed improved in counting. We note that TallyQA (test) only includes very few examples with high numbers ($\geq 9$), leading to larger uncertainties in the reported results. As we discuss in this work, all evaluated text-to-image models already struggle with much smaller numbers ($\geq 4$), and similar issues are observed with VLMs. We speculate that as models continue to improve rapidly, benchmarks with high-quality images containing higher counts of objects will become even more important in model development and evaluation.

### D.2.2 Fine-tuning PaLIGemma on GECKONUM

We briefly investigate the utility of GECKONUM as synthetic training data. For this we fine-tune PaLIGemma on different mixtures of TallyQA (train) and Imagen A+B+C+D, and evaluate these on TallyQA (test) and the split of Muse-B images, see Tables A12 and A13. We have 53.053 Imagen-A+B+C+D images in total, which we can use for fine-tuning, while the size of TallyQA (train) is $249, 318$. We perform two sets of experiments:

- **Experiment 1**, in which we match GECKONUM data roughly 1:1 with TallyQA data to explore the utility of the 53k synthetic data points compared to 53k real data points.
- **Experiment 2**, in which we train on all available TallyQA data and add GECKONUM data to explore the benefits of enlarging an established training set with synthetic data.

In **Experiment 1** we control for fine-tuning set sizes and pair the 53k Imagen images with 53k randomly sampled TallyQA train images (for a total of 106k examples) and compare them to fine-tuning on only 106k randomly sampled TallyQA train images. We also consider paring them with half the amount of TallyQA images (26k) as well as only training on the Imagen images. To control

for the fact that Imagen and Muse share the same object classes, we also consider a split of Imagen and Muse into two disjoint object sets of "frequent" and "rare" as described above and only train on the "frequent" subset and treating the "rare" subset as a held-out set.

The results in Table A12 can be summarized as:

- Including GECKONUM Imagen images *in addition to* TallyQA images neither boosts nor deteriorates performance on TallyQA (test) significantly; there is a slight improvement on TallyQA (test) "simple" and a slight deterioration on TallyQA (test) "complex".

- However, *replacing* some or all of the TallyQA (train) images with Imagen images does deteriorate performance on TallyQA (test) by up to a few percentage points.

- Including Imagen images always markedly improves performance on Muse-B (by more than 25 p.p.). This is still true when only training on "frequent" classes while evaluating only on "rare" classes, though the improvement is slightly smaller in this case (by about 20 p.p.). There is clear generalization from frequent to rare classes.

- Fine-tuning on Imagen only already leads to strong performance on TallyQA (test) "simple" (about 70% accuracy compared to 35% for the base model and 82% when training on TallyQA) but not "complex" (only an increase to 44.9% from 37.8% compared to 67.8% when training on TallyQA). The latter makes intuitive sense as the GECKONUM VQA questions resemble "simple" TallyQA questions (How many <object> are there in the image?) but not "complex" ones (How many <object>s have <property>?, e.g. "How many giraffes are lying down?").

Thus, as noted in Section 6, adding Imagen images to the training mix does not hurt performance on TallyQA (test), but it allows us to achieve much better results on other data splits not covered by the TallyQA training data as well (the Muse-B split in this case). We also found that during fine-tuning on TallyQA (train) the model slightly overfits to the TallyQA dataset in that the best TallyQA (test) performance is achieved at the end of fine-tuning while the best performance on Muse-B is achieved partway through fine-tuning and performance reduces slightly as fine-tuning continues.

In **Experiment 2**, we use the full TallyQA (train) set and add the 53k Imagen images in addition, and compare this mixture to slight variations where we account for the number of training examples (by removing 53k examples from TallyQA) and the number of training steps (by training on TallyQA for longer). The results are shown in Table A13. We replicate previous results and observe the same trends; the changes on TallyQA (test) performance are very small ($< 1\%$ in the worst case but usually much smaller) when including Imagen images while performance on Muse-B is drastically improved even in the held-out case.

Table A12: Accuracy of PaLIGemma fine-tuned on different data mixtures (at resolution $448 \times 448$ pixels). Here, we only train on a subset of TallyQA (train), which in total has approximately 250k examples. Imagen here means Imagen A+B+C+D. We provide the approximate number of unique data points in brackets.

| Fine-tuning data | TallyQA test | | Muse-B | | |
|---|---|---|---|---|---|
| | simple | complex | all | freq. | rare |
| TallyQA (106k) | 83.1 | 69.4 | 70.3 | 76.5 | 63.2 |
| TallyQA (53k) | 82.0 | 67.8 | 68.3 | 75.0 | 60.5 |
| TallyQA (53k) + Imagen (freq & rare, 53k) | 82.4 | 67.8 | 89.4 | 90.0 | 88.8 |
| TallyQA (53k) + Imagen (only freq, 28k) | 82.2 | 68.3 | 86.0 | 89.5 | 82.0 |
| TallyQA (26k) | 81.1 | 66.5 | 68.7 | 75.2 | 61.3 |
| TallyQA (26k) + Imagen (freq & rare, 53k) | 81.5 | 66.4 | 89.6 | 90.0 | 89.2 |
| Imagen (freq & rare, 53k) | 70.7 | 44.9 | 90.0 | 90.1 | 89.7 |
| Imagen (only freq, 28k) | 70.1 | 44.7 | 85.0 | 88.6 | 80.7 |
| Base model (no fine-tuning) | 35.4 | 37.8 | 67.0 | 71.0 | 62.4 |

Table A13: Accuracy (in %) of PaLIGemma fine-tuned on different data mixtures (at resolution $448 \times 448$ pixels). Here, we train on all TallyQA (train) images. Imagen here means Imagen-A+B+C+D and "all" refers to freq. and rareuent objects.

| Fine-tuning data | data points | TallyQA test simple | complex | Muse-B all | only freq. | rare |
|---|---|---|---|---|---|---|
| TallyQA | 250k | 84.7 | 72.8 | 71.1 | 76.6 | 64.8 |
| TallyQA (only 200k) + Imagen (all) | 250k | 84.7 | 71.9 | 90.6 | 90.9 | 90.3 |
| TallyQA + Imagen (all) | 300k | 84.9 | 72.7 | 90.5 | 91.2 | 89.8 |
| TallyQA + Imagen (only freq) | 275k | 84.9 | 72.5 | 87.3 | 90.4 | 83.6 |
| TallyQA (longer training) | 250k | 84.7 | 72.4 | 69.8 | 76.9 | 61.7 |
| Base model (no fine-tuning) | – | 35.4 | 37.8 | 67.0 | 71.0 | 62.4 |