# OpenReview forum: "Evaluating Numerical Reasoning in Text-to-Image Models"
_NeurIPS.cc/2024/Datasets_and_Benchmarks_Track — NeurIPS 2024 Track Datasets and Benchmarks Poster_

### Official Review · Reviewer_m6se · 2024-07-24

**Rating:** 7
**Confidence:** 4
**Correctness:** Yes
**Clarity:** Yes

**Review:**

Weakness:

1. More open-source or closed-source T2I models should be included in the evaluation and benchmarking, i.e. stable diffusion v3, Meta Chamelon, etc. For LLM-based T2I models, the multimodal in-context learning image generation can be also considered as an evaluation pattern to check whether such models can follow the numerical demonstrations in the context.

2. Some correlation between better numerical reasoning and human preference can be also investigated. You can evaluate the generated images with some model-based metrics (LLMScore, GPT4V-Eval) or automatic metrics to check whether there is positive correlation between better numerical reasoning and better quality score for the generated images.

**Strengths:**

1. The numerical reasoning is a huge challenge for current SOTA T2I models and there is no effective benchmark on evaluating this perspective of T2I models. The previous work Gecko mentioned this as a perspective in evaluating T2I models but without proposing a useful benchmark.

2. The benchmark's coverage and diversity in prompts are remarkable and robust, making the T2I models hard to overfit to this benchmark.

**Additional Feedback:**

None

**Documentation:**

Yes

**Limitations:**

Yes

**Opportunities For Improvement:**

Refer to weakness.

**Relation To Prior Work:**

Yes

**Summary And Contributions:**

This work proposed GeckoNum, a novel benchmark for evaluating the numerical capability of T2I models. This research problem is of high value as current T2I models often generates wrong number of objects in the generated image. It is a long challenge in developing T2I models. This benchmark focuses on three aspects of numerical reasoning, the exact number, the linguistic approximate number, and the conceptual quantitive number. The designed subtasks for numerical reasoning are aligned with real-user prompts and will motivate the promotion of numerical reasoning of T2I models.

---

> ### Author Rebuttal · Authors · 2024-08-16
>
> We appreciate your suggestions that aim to improve the quality of our work, and would like to ask for some clarifications to address your concerns.
>
> > More open-source or closed source T2I models.
>
> As mentioned in our response to Reviewer ra8a, we will be adding evaluation results from five more open- and closed-source models: Stable Diffusion (SD) 1.5, SD 2.1, SDXL, SD 3 and Midjourney 6 to address concerns regarding comprehensiveness. We are in the process of generating the data, and we will get back with results as soon as we have them.
>
> > You can evaluate the generated images with some model-based metrics (LLMScore, GPT4V-Eval) or automatic metrics to check whether there is positive correlation between better numerical reasoning and better quality score for the generated images.
>
> In Section 6 “Evaluation of auto-eval metrics” we found that many auto-eval metrics that evaluate text-to-image alignment poorly capture counting. Specifically, VNLI is the only metric that reliably distinguishes between correct and incorrect images with respect to object counts. Other auto-metrics we test (TIFA, DSG, Gecko) are similar to LLMScore insofar that they all rely on LLMs in evaluation, and we show that they frequently fail to discriminate between images.
>
> As we are unsure whether we correctly interpreted your question, we would also like to ask for a further clarification of what is meant by “better quality score”. For example, are you suggesting we explore whether models that generate higher quality images (as measured by some metric of image quality) are also better at numerical reasoning?

---

> > ### Comment · Reviewer_m6se · 2024-08-17
> >
> > Thanks for the responses. Will keep the score.

---

### Official Review · Reviewer_ra8a · 2024-07-25
**A benchmark to evaluate if text-to-image models can generate contents with given numerical descriptions**

**Rating:** 5
**Confidence:** 4
**Correctness:** The benchmark in the submission is de…
**Clarity:** The writing quality could be further …

**Review:**

Quality: This paper focuses on the evaluation of the numerical reasoning of text-to-image models and whether the models can generate images where the number of objects aligns with the prompts. The paper proposes three tasks: exact numbers, approximate numbers, and conceptual reasoning, and uses 12 templates for prompts in total as a benchmark.

Clarity: This paper is easy to follow, but the writing quality needs further improvement. It would be better if the authors could introduce the whole procedure of numerical evaluation after introducing the tasks.

Originality: The originality of the newly proposed benchmark with three numerical tasks is good.

Significance: Failing to reflect the number of objects is a common shortcoming of text-to-image models and the idea of using the proposed three numerical tasks is intuitive and effective in evaluating the numerical reasoning of text-to-image models. However, the requirement of human annotations might limit the usage of the benchmark.

**Strengths:**

1. This paper focuses on the evaluation of numerical reasoning of text-to-image models, which is a task that text-to-image models often fail on.

2. The design of three numerical tasks is reasonable and intuitive, inspired by the way how children learn numbers.

3. The experiments are comprehensive with a detailed analysis of different factors, such as whether using digits for numbers, or the noun choice in the prompts.

**Additional Feedback:**

No additional feedback.

**Documentation:**

The paper provides sufficient details for the proposed benchmark. The benchmark link is not available yet.

**Ethics:**

There are no or minor ethical concerns with the submission.

**Limitations:**

The authors adequately discussed the limitations of their work.

**Opportunities For Improvement:**

1. The comparison between the performance of models and the baseline is appreciated, but the design of the baselines does not fully convince me. Is there any better way to use it as the baseline?

2. The experiments discussed the impacts of different factors, such as using digits or words for numbers or using frequent or non-frequent words as nouns. Given the results, did the curation of the prompts in the benchmark consider such impacts? Did the authors balance different cases?

3. Will other attributes (such as colors) influence the generation or evaluation of the images? It seems that the attributes are only considered in the first task, while absent in the remaining two tasks.

**Relation To Prior Work:**

This paper discussed prior work in detail.

**Summary And Contributions:**

This paper evaluates a range of text-to-image models on numerical reasoning tasks of varying difficulties and proposes a benchmark of three numerical tasks, with 1386 text prompts to evaluate the performance of numerical reasoning. The results showed that the current text-to-image models have the basic ability of numerical reasoning and exhibited accurate performance with small exact quantities. However, the models have difficulties with approximate numbers or conceptual reasoning.

---

> ### Author Rebuttal · Authors · 2024-08-16
>
> We are grateful for the detailed feedback and would like to provide additional explanations to address some of your questions. We would also appreciate your suggestions for improvement.
>
>
> >  Is there any better way to use it as the baseline?
>
> We realize the term "baseline" may have been misleading in this context. What we want to convey is: "how difficult is this numerical task?" and we use the performance of randomly guessing answers (i.e. $1/(num\ possible\ answers)$) together with model performance  as a proxy for this. This "baseline" assumes we either pick numbers from a range of answers or from a multiple choice question. We will make this clearer in the paper. We can use the term “random chance performance” instead of baseline if you believe this would improve clarity.
>
> In general, "True" baselines for evaluation of text-to-image models are non-trivial given the large space of possible image outputs that may, in the worst case, be completely unrelated to the question. We welcome your ideas on what you would consider to be a better baseline in this case.
>
> > The experiments discussed the impacts of different factors, such as using digits or words for numbers or using frequent or non-frequent words as nouns. Given the results, did the curation of the prompts in the benchmark consider such impacts? Did the authors balance different cases?
>
> Yes, we were deliberate in the design of prompts to sufficiently cover different cases. Specifically, in the Appendix Sections A.1 “Words and Word Frequencies” we have listed all words, including coverage per frequency type, and in A.2 “The Distribution of Prompt Types in the Benchmark” we listed the breakdown of number prompts per prompt type. We also highlight that for all tasks, we sample five different images per prompt for each model.
>
> > Will other attributes (such as colors) influence the generation or evaluation of the images? It seems that the attributes are only considered in the first task, while absent in the remaining two tasks.
>
> Due to combinatorial explosion it would be difficult to study every possible combination of “number x noun x attribute”, however we did cover some additional attributes such as size and color in Task 3, for example:
>
> `A pencil broken into two pieces. One piece is twice the size of the other.`
>
> `An image of a pencil where one half of it is orange and the other half is black.`
>
> `An image of a banana where one half of it is red and the other half is blue.`
>
> We also discussed more general observations and challenges in evaluation of color in C.3.1 and C.3.3 in the Appendix, where we looked at qualitative examples. We prioritized coverage of numerical concepts in Tasks 2 and 3 given that some of those prompts can be difficult even without additional attributes, and given that color terms did not significantly impact model accuracy in Task 1 (Tables 9 and 10 in Appendix B.1.4).

---

> > ### Comment · Reviewer_ra8a · 2024-08-26
> >
> > Thanks for the detailed rebuttal from the authors. I agree that it's impossible to study every combination of numbers, nouns, and attributes. Since this benchmark focuses on the evaluation of numerical reasoning, it makes sense not to use attributes in tasks. For task 3 integrated with attributes, will introducing attributes have any negative influence on the numerical reasoning of models? Besides, I would like to ask the authors' opinion on the limitation mentioned in the global rebuttal, which I failed to find it in the attached PDF.

---

> > > ### Author Rebuttal · Authors · 2024-08-27
> > >
> > > > For task 3 integrated with attributes, will introducing attributes have any negative influence on the numerical reasoning of models?
> > >
> > > Thank you for the interesting question -- to answer it, we divided the relevant prompts in Task 3 into two groups:
> > >
> > > Group A: Prompts that contain only a fractional quantity, without color terms (eg `There is one apple and half of another apple on the table.`)
> > >
> > > Group B: Prompts that contain both a fractional quantity and a color term (eg `An image of a pencil where one half of it is red and the other half is blue.`)
> > >
> > > Then, we conducted the Mann–Whitney U test ($\alpha=.05$) to compare differences in accuracy distribution between these two groups. The results showing means, standard error of the means, test statistic and p value:
> > >
> > > |    | model         |   Mean-GroupA |   SEM-GroupA |   N1 |   Mean-GroupB |   SEM-GroupB |   N2 |   stat |     p |
> > > |---:|:--------------|--------------:|-------------:|-----:|--------------:|-------------:|-----:|-------:|------:|
> > > |  0 | DALLE 3       |         40.22 |         3.12 |  40  |         47.32 |         4.85 |  20  |  311   | 0.163 |
> > > |  1 | Imagen-A      |         41.86 |         3.62 |  35  |         15.82 |         2.29 |  20  |  638   | 0     |
> > > |  2 | Imagen-B      |         41.18 |         2.98 |  35  |         12.62 |         2.31 |  20  |  679   | 0     |
> > > |  3 | Imagen-C      |         48.32 |         3.24 |  35  |         23.73 |         3.1  |  20  |  611   | 0     |
> > > |  4 | Imagen-D      |         48.42 |         3.88 |  40  |         21.02 |         0.79 |  20  |  751   | 0     |
> > > |  5 | Midjourney v6 |         49.04 |         3.6  |  40  |         31.45 |         4.06 |  20  |  615.5 | 0.001 |
> > > |  6 | Muse-A        |         46.5  |         3.36 |  35  |         23.32 |         3.98 |  20  |  611.5 | 0     |
> > > |  7 | Muse-B        |         49.73 |         2.95 |  40  |         21.53 |         3.95 |  20  |  691.5 | 0     |
> > > |  8 | SD1.5         |         47.62 |         2.72 |  39  |         27.26 |         3.25 |  19  |  640.5 | 0     |
> > > |  9 | SD2.1         |         45.57 |         2.69 |  40  |         22.73 |         2.27 |  20  |  703   | 0     |
> > > | 10 | SD3           |         42.56 |         2.01 |  40  |         32.63 |         4.16 |  20  |  561   | 0.011 |
> > > | 11 | SDXL          |         42.78 |         2.05 |  39  |         32.67 |         4.56 |  20  |  572   | 0.004 |
> > >
> > > Based on the results, it does seem that 11/12 models will be significantly less accurate when color terms are combined with more complex numerical concepts such as fractions. In other words -- yes, introducing attributes will have negative influence on numerical reasoning. The interesting exception is again Dalle 3, the model where there was no significant difference and that consistently appears to be the most robust across all our experiments. Connecting this to findings in Task 1, these results seem to indicate that when the color terms appear in simpler numerical tasks, models' performance is less affected compared to more complex numerical tasks.
> > >
> > > While we find this to be an interesting finding, we note that these conclusions are based on a smaller prompt set where we have not directly controlled for different combinations of colors and nouns between groups. We nonetheless believe the results are valuable as a stepping stone for further research on the interplay of compositional reasoning skills, and will add them in the paper.
> > >
> > > > Besides, I would like to ask the authors' opinion on the limitation mentioned in the global rebuttal, which I failed to find it in the attached PDF.
> > >
> > > In following the rebuttal guidelines (`(should be one page and contain only Figures and Tables)`) we did not see an obvious way to add the whole revised paper with highlighted changes in OpenReview, so we apologise for the lack of clarity on our side. We hope that adding the whole paragraph here, as we currently have it in the revised paper,
> > >  addresses your comment:
> > >
> > > > Our approach relies on human annotators to provide counts of objects in an image, as is often the gold standard in evaluation of text-to-image models. However, such annotation processes are laborious, costly and do not scale. We expect that the rapid improvement of pipelines and frameworks for evaluation of text-image alignment [10, 35, 13] will reduce the need for manual annotation.  In addition, we identified three important numerical capabilities  for which  we  have  manually  designed human annotation templates, but numerical cognition in humans spans a wide set of gradually developing capabilities that might require additional evaluation templates. As discussed in Section 5.3, the design of protocols for evaluation of more complex aspects of numerical reasoning remains an important open challenge with rapid advancements in models.
> > >
> > > > References: [10] J. Cho, A. Zala, and M. Bansal. Visual programming for step-by-step text-to-image generation and evaluation. Advances in Neural Information Processing Systems, 37, 2023, [13] T. Gokhale, H. Palangi, B. Nushi, V. Vineet, E. Horvitz, E. Kamar, C. Baral, and Y. Yang. Bench-marking spatial relationships in text-to-image generation.arXiv preprint arXiv:2212.10015, 2022, [35] M. Yarom, Y. Bitton, S. Changpinyo, R. Aharoni, J. Herzig, O. Lang, E. Ofek, and I. Szpektor.What you see is what you read?  improving text-image alignment evaluation.Advances in Neural Information Processing Systems, 37, 2023

---

### Official Review · Reviewer_qpir · 2024-07-25
**An interesting study with in-depth analysis on a specific topic**

**Rating:** 6
**Confidence:** 3
**Correctness:** The authors’ claims appear to be valid.
**Clarity:** The paper is well written.

**Review:**

**Pros**

- The benchmark dataset is composed of a variety of numeric expressions that can be articulated in language, enabling the verification of T2I models’ numerical reasoning abilities.
- Through analysis, the authors effectively highlight the numerical reasoning characteristics of each T2I model and the features of existing auto-evaluation metrics/models.
- The dataset, GEKONUM, can be used to assess the counting capabilities of Vision-Language Models (VLMs). The study demonstrates that using GEKONUM prompts and synthetic images generated by Imagen for fine-tuning can enhance VLMs’ counting abilities.


**Cons**
- The prompt composition in Task 1 is overly simplistic. Given that the generation models may be biased towards certain words, it seems necessary to test with a broader range of words beyond the ones used (apple, cat, koala, mushroom, and bottle).
- To improve T2I models’ text alignment, particularly in spatial relations and counting, many pipelines utilizing base T2I models and large language models (LLMs) have been proposed. It would have been beneficial to include human evaluation results and analyses showing whether these pipelines actually improve counting and spatial relation generation abilities.
- It would be beneficial to include a more detailed analysis of the reasons why each T2I model failed in specific tasks or sub-tasks.

**Strengths:**

The significance of this work lies in proposing a benchmark that focuses on numerical reasoning, among the many benchmarks that evaluate the generative capabilities of T2I models. By analyzing how T2I models respond to various numeric expressions, this benchmark can contribute to their improvement and development.

**Additional Feedback:**

None

**Documentation:**

Sufficient details about the proposed dataset and experiments are provided in the appendices.

**Limitations:**

The authors haven't addressed the limitations of their work. However, it appears that there are no potential negative societal impacts.

**Opportunities For Improvement:**

Please refer to the Cons of the Review section for further details.

**Relation To Prior Work:**

The authors explained how their work differs from previous T2I evaluation benchmarks.

**Summary And Contributions:**

The authors propose a benchmark dataset called GEKONUM to measure the numerical reasoning performance of text-to-image (T2I) generation models. The dataset includes numerical expression prompts categorized into 3 tasks and 12 types, capturing a wide range of numerical expressions. Using this proposed dataset, the numerical reasoning abilities of well-known T2I models are validated through human evaluation. Additionally, the authors verify the effectiveness of an auto-evaluation metric using the proposed GEKONUM in assessing the generative capabilities of T2I models.

---

> ### Author Rebuttal · Authors · 2024-08-16
>
> Thank you for your comments. We would like to clarify some aspects of our work, and would like to ask you for more details to be able to adequately address concerns.
>
>
> > The prompt composition in Task 1 is overly simplistic. [...] it seems necessary to test with a broader range of words beyond the ones used (apple, cat, koala, mushroom, and bottle).
>
> We have indeed tested a wider range of words (40 in total) and will clarify this in the paper.  Specifically, the detailed list of all words in Task 1 is available in Appendix A.1, while Table 1 in the main paper body shows a small preview of prompts. Table 2 indeed includes performance on all words on Task 1. We assume you wrote your comment with Figure 3 (and L166) in mind, where we selected a smaller subset of words that exist in all prompt types. Here, to enable fair comparison between models on different prompt structures, we decided to keep the words constant (hence the small set of words) and only vary the prompt structure. We focused on this subset to be able to collect more data across various models, and as it already highlights the shortcoming of these models. We do, however, acknowledge your concern that this may reflect bias in models, and if you believe that including all words in Fig. 3 would constitute a fairer comparison we are ready to replace the figure in question.
>
>
> > It would have been beneficial to include human evaluation results and analyses showing whether these pipelines actually improve counting and spatial relation generation abilities.
>
> Could you please clarify which specific pipelines you mean?
>
> We agree that evaluation of such methods, and relatedly, auto-eval metrics is an important subject to advance state-of-the-art models in image generation. Although this was not our main goal, we touched upon this topic in Section 6 “Evaluation of auto-eval metrics”, where we found that many auto-eval metrics are unable to capture counting. While human evaluation remains the gold standard, we hope that our findings and the benchmark further incentivize research to improve auto-metrics, especially for tasks you mention, as to reduce the need for human evaluation. Such improved auto-metrics would make it easier for a broader research community to evaluate models and pipelines.
>
>
> > It would be beneficial to include a more detailed analysis of the reasons why each T2I model failed in specific tasks or sub-tasks.
>
> We discuss some model specific failures in the Appendix (B.2 Qualitative Analysis of Model Failures), where we attribute certain kinds of failures to older versions of models (ie. models with fewer parameters). If you believe this warrants a place in the main paper, we are happy to add it. Also, please let us know if you have suggestions for additional analyses on this data. While we focused on the “how” question of model failure, answering “why” for each model might require extensive ablations (such as those done in [Ref1]), which is not a research question we intended to answer in this paper.
>
>
> > The authors haven't addressed the limitations of their work.
>
> In Appendix C.3 we highlight the limitations of our methodology, specifically for Task 3, however we recognize that a dedicated paragraph in the main paper would be useful and we will amend the main paper to reflect this.
>
>
> References:
> Ref 1: Vasconcelos et al (2024): Greedy Growing Enables High-Resolution Pixel-Based Diffusion Models, [pdf](https://arxiv.org/pdf/2405.16759)

---

> > ### Author Rebuttal · Authors · 2024-08-22
> >
> > As we have expanded the paper with new results (attached here for your convenience), we would also like to highlight that we have added a paragraph dedicated to addressing limitations of our work in the final section of the paper, specifically mentioning the reliance of our method on human annotators and annotation templates that test for a fixed set of numerical reasoning skills. If there are any outstanding concerns you would like to see addressed, please let us know.

---

### Official Review · Reviewer_XZvx · 2024-07-25
**Evaluating Numerical Reasoning in Text-to-Image Models**

**Rating:** 7
**Confidence:** 5
**Correctness:** Yes
**Clarity:** Yes

**Review:**

Strength:
The study provides a thorough evaluation of text-to-image generative models on numerical reasoning tasks, highlighting their limitations in a detailed manner. The creation of the GeckoNum benchmark is a significant contribution.

Weakness:
The paper primarily focuses on editing prompts for numerical reasoning tasks using three text-to-image generative models: DALL-E, Muse, and Imagen. However, it lacks a deep analysis of the underlying factors contributing to the models’ performance and limitations. While the paper designs various forms of numerical reasoning tasks to validate the limitations of existing methods, it does not provide sufficient insights or theoretical considerations on how to improve these tasks.
The paper does not analyze how different model architectures, training methodologies, and variations in training data contribute to the observed limitations in numerical reasoning tasks. Additionally, the study does not test a wide range of open-source models and datasets, lacking a comprehensive evaluation. This omission prevents a thorough understanding and generalization of the findings, and the paper fails to provide readers with standout insights or summarizations on overcoming these limitations.

**Strengths:**

The study provides a thorough evaluation of text-to-image generative models on numerical reasoning tasks, highlighting their limitations in a detailed manner. The creation of the GeckoNum benchmark is a significant contribution.

**Additional Feedback:**

Providing more conclusive and generalized summaries could help demonstrate the effectiveness of the benchmarks.

**Documentation:**

Yes

**Limitations:**

Yes

**Opportunities For Improvement:**

The paper primarily focuses on editing prompts for numerical reasoning tasks using three text-to-image generative models: DALL-E, Muse, and Imagen. However, it lacks a deep analysis of the underlying factors contributing to the models’ performance and limitations. While the paper designs various forms of numerical reasoning tasks to validate the limitations of existing methods, it does not provide sufficient insights or theoretical considerations on how to improve these tasks.
The paper does not analyze how different model architectures, training methodologies, and variations in training data contribute to the observed limitations in numerical reasoning tasks. Additionally, the study does not test a wide range of open-source models and datasets, lacking a comprehensive evaluation. This omission prevents a thorough understanding and generalization of the findings, and the paper fails to provide readers with standout insights or summarizations on overcoming these limitations.

**Relation To Prior Work:**

Yes

**Summary And Contributions:**

The paper presents a comprehensive evaluation of text-to-image generative models, focusing on their numerical reasoning capabilities. The authors find that even the most advanced models struggle with generating an exact number of objects beyond small quantities. The models’ performance is highly context-dependent and deteriorates with increasing numbers. Additionally, the models show a poor understanding of linguistic quantifiers, the concept of zero, and advanced concepts such as fractional representations. To address these issues, the authors introduce GeckoNum, a novel benchmark that includes prompts, generated images, and human annotations to evaluate numerical reasoning in text-to-image models.

---

> ### Author Rebuttal · Authors · 2024-08-16
>
> Thank you for your review. While we agree that experiments you suggest would provide a comprehensive explanation of different factors that affect model performance, these experiments answer a different question than what this paper studies: these experiments shed light on *what contributes to performance* on numerical reasoning while we ask how we can *evaluate* numerical reasoning. In particular, the primary focus of our work is extensive model evaluation, evaluation protocol design and dataset curation for numerical reasoning. As such, the suggested model ablations are not in the scope of this work. Moreover, such experiments often examine one model family in-depth, while we focus on comparing various models.  We note a recent paper, where various aspects of a specific model’s architecture (ie. Imagen Vermeer) were ablated [Ref1]. For counting (ie “exact number generation”), it was shown that the accuracy on the task increased with model size (Fig 4). We will amend our manuscript to include this reference, as we agree this is indeed an important subject that should be further investigated in future work.
>
> As a result, we focus our ablations on comparing models, and challenge the notion that we do *”not analyze how different model architectures, training methodologies, and variations in training data contribute to observed limitations in numerical reasoning"* --- the three model families in the paper are representative of different model architectures, covering pixel-based (Imagen) and latent-based models (Muse, Dall-e). Furthermore, within some families (ie Imagen, Muse) we report results on models covering different generations (ie. earlier models had fewer parameters, and different  datasets). To make it more clear in the paper that these are different architectures, we will expand the section on technical details for models and mention model parameter sizes including data the models were trained on. In addition, we will be adding evaluation results from five more models, Stable Diffusion (SD) 1.5, SD 2.1, SDXL, SD 3 and Midjourney 6 to address concerns regarding comprehensiveness. We will get back to you as quickly as possible after we have the results.
>
> References:
>
> Ref1: Vasconcelos et al (2024): Greedy Growing Enables High-Resolution Pixel-Based Diffusion Models ([pdf](https://arxiv.org/pdf/2405.16759))

---

> > ### Author Rebuttal · Authors · 2024-08-22
> >
> > Following up on our previous response, we have now incorporated new results in the paper with five new models (SD1.5, SD2.1, SDXL, SD3 and Midjourney 6), to also include open-source models. The overview of results is available in the attached 1-page PDF. Additional results reinforce our existing findings, and also provide novel insights:
> > * Numerical accuracy of models is trivially affected by manipulations we investigate. For example, 10/12 models are significantly less accurate when numbers are represented as digits as opposed to words, confirming robustness of our methods and observations
> > * While Dalle-3 remains the strongest model on numerical tasks, SD3 and Midjourney v6, its contenders on Task 1, remain vulnerable to many such manipulations, showing that robust numerical reasoning is an open challenge even for the most recent models
> >
> > We have expanded the section on technical details and added model parameter sizes and training data information for Imagen models to facilitate comparison of models.
> >
> > Finally, with a total of 801K human annotations, 87K generated images, 12 open-source and proprietary state-of-art models, rigorous and extensive statistical analyses, we hope that you will find our evaluations conclusive, general and useful in guiding the improvement of numerical reasoning skills in generative models.

---

> > > ### Comment · Reviewer_XZvx · 2024-08-26
> > >
> > > Thank you for your reply. It has addressed some of my concerns, and I will increase my score. One more question: Is there a URL (as mentioned in the submission instructions for this track) to publicly showcase your dataset or evaluation benchmark? This is very important in the dataset and benchmarks track, as it helps others with reproducibility.

---

> > > > ### Author Rebuttal · Authors · 2024-08-27
> > > >
> > > > Thank you. We are glad we were able to address some of your concerns, and welcome any further feedback that you believe would improve our research.
> > > >
> > > > > Is there a URL (as mentioned in the submission instructions for this track) to publicly showcase your dataset or evaluation benchmark?
> > > >
> > > > Yes, the URL is: https://github.com/google-deepmind/geckonum_benchmark_t2i/. The repository contains the benchmark, documentation of the format and [a preview of the dataset](https://github.com/google-deepmind/geckonum_benchmark_t2i/blob/main/examples.md) for each numerical task. We have updated the paper to include the URL.

---

### Author Response · Authors · 2024-08-16

We thank all reviewers for comments and helpful suggestions that aim to improve our work. We have individually addressed your comments under respective reviews, including several requests for clarification where we would appreciate further feedback while we run additional experiments.

---

> ### Author Rebuttal · Authors · 2024-08-22
>
> We're following up on our previous comment to report that we have updated the paper to incorporate new results after including suggestions made by reviewers. The summary of changes we made is listed here, and, where relevant, more details are provided under respective comments:
> * Added results for 5 additional models (SD1.5, SD2.1, SDXL, SD3, and Midjourney v6). The results confirm the robustness of our evaluation approach, and validity of our findings regarding weaknesses of numerical cognition of text-to-image models
> * Included additional technical details about different model architectures and their training, where applicable
> * Added a paragraph on limitations in the final section
> * Additional clarifications to improve writing clarity, and added missing references to related work
>
> The overview that includes new results is available in the attached 1-page PDF.  Please let us know if there are any outstanding comments we may have not addressed. Thank you.

---

### Decision · Program_Chairs · 2024-09-26

**Decision:**

Accept (Poster)

**Comment:**

This paper introduces GECKONUM, a comprehensive benchmark designed to evaluate the numerical reasoning capabilities of text-to-image generative models. The authors investigate a variety of models, including DALL-E 3, Imagen, and Muse, across multiple numerical reasoning tasks, such as exact number generation, approximate number reasoning, and conceptual quantitative reasoning. With over 48,000 generated images and nearly 500,000 human annotations, the benchmark offers a thorough evaluation of how well these models handle numerically specified prompts, revealing that most models exhibit limitations in accurately depicting numbers.

Reviewers generally praised the paper for addressing an important and underexplored area of text-to-image model evaluation. One reviewer highlighted the paper’s significance in advancing evaluations of numerical reasoning, describing the GECKONUM benchmark as a significant contribution to the field. Reviewers also commended the breadth and depth of the benchmark, which effectively highlights key weaknesses in current models.

However, there were also some concerns. One reviewer pointed out that the paper could have expanded its analysis by proposing methods for improving numerical reasoning, rather than focusing solely on evaluation. Another reviewer expressed a desire for the paper to explore the underlying factors contributing to model performance, such as the impact of different architectures or training methodologies. The authors addressed some of these concerns in their rebuttal, adding new results from additional models (e.g., Stable Diffusion, Midjourney) and clarifying the scope of their work, which focuses on evaluation rather than model improvement.

Despite these concerns, the consensus among the reviewers is that the paper provides a valuable and thorough evaluation that will benefit the research community. The addition of new models further strengthens the robustness of the benchmark, and the detailed evaluation methodology offers a significant resource for future studies.